# Learning Graph Structure from Convolutional Mixtures

**Max Wasserman**                                                    *mwasser6@cs.rochester.edu*
*Department of Computer Science*
*University of Rochester*

**Saurabh Sihag**                                              *sihags@pennmedicine.upenn.edu*
*Department of Electrical and Systems Engineering*
*University of Pennsylvania*

**Gonzalo Mateos**                                                 *gmateosb@ece.rochester.edu*
*Department of Electrical and Computer Engineering*
*University of Rochester*

**Alejandro Ribeiro**                                                 *aribeiro@seas.upenn.edu*
*Department of Electrical and Systems Engineering*
*University of Pennsylvania*

**Reviewed on OpenReview:** *https://openreview.net/forum?id=OILbPOWErR*

## Abstract

Machine learning frameworks such as graph neural networks typically rely on a given, fixed graph to exploit relational inductive biases and thus effectively learn from network data. However, when said graphs are (partially) unobserved, noisy, or dynamic, the problem of inferring graph structure from data becomes relevant. In this paper, we postulate a graph convolutional relationship between the observed and latent graphs, and formulate the graph structure learning task as a network inverse (deconvolution) problem. In lieu of eigendecomposition-based spectral methods or iterative optimization solutions, we unroll and truncate proximal gradient iterations to arrive at a parameterized neural network architecture that we call a Graph Deconvolution Network (GDN). GDNs can learn a distribution of graphs in a supervised fashion, perform link prediction or edge-weight regression tasks by adapting the loss function, and they are inherently inductive as well as node permutation equivariant. We corroborate GDN's superior graph learning performance and its generalization to larger graphs using synthetic data in supervised settings. Moreover, we demonstrate the robustness and representation power of GDNs on real world neuroimaging and social network datasets.

## 1 Introduction

Inferring graphs from data to uncover latent complex information structure is a timely challenge for geometric deep learning (Bronstein et al., 2017) and graph signal processing (GSP) (Ortega et al., 2018). But it is also an opportunity, since network topology inference advances (Dong et al., 2019; Mateos et al., 2019; Giannakis et al., 2018) could facilitate adoption of graph neural networks (GNNs) even when no input graph is available, e.g. for disease prediction in healthcare settings (Kazi et al., 2023), or, 3D point cloud segmentation in computer graphics (Wang et al., 2019). The problem is also relevant when a given graph is too noisy or perturbed beyond what stable (possibly pre-trained) GNN architectures can effectively handle (Gama et al., 2020a). Early empirical evidence suggests that even when a graph is available, the structure could be further optimized for a downstream task e.g., when parameterizing a GNN used for node classification in citation, social, and protein-protein interaction networks (Zhao et al., 2021). Sometimes the observed graph's large edge set can make a downstream task computationally expensive, which motivates seeking a sparser graph which retains properties of the original one – thus making the task feasible (Spielman & Srivastava, 2011).

In this paper, we posit a convolutional model relating observed and latent undirected graphs and formulate the graph structure learning task as a supervised network inverse problem; see Section 2 for a formal problem statement. This fairly general model is motivated by various practical domains outlined in Section 3, such as identifying the structure of network diffusion processes (Segarra et al., 2017; Pasdeloup et al., 2018), as well as network deconvolution and denoising (Feizi et al., 2013). We propose a parameterized neural network (NN) model, termed graph deconvolution network (GDN), which we train in a supervised fashion to learn the distribution of latent graphs. The node permutation equivariant architecture is derived from the principle of algorithm unrolling used to learn fast approximate solutions to inverse problems (Gregor & LeCun, 2010; Sprechmann et al., 2015; Monga et al., 2021), an idea that is yet to be fully explored for graph structure identification. Since layers directly operate on, combine, and refine graph objects (instead of nodal features), GDNs are inherently inductive and can generalize to graphs of different size. This allows the transfer of learning on small graphs to unseen larger graphs, which has significant implications in domains like social networks and molecular biology (Yehudai et al., 2021). Our experiments demonstrate that GDNs are versatile to accommodate link prediction or edge-weight regression aspects of learning the graph structure, and achieve superior performance over various competing alternatives. Building on recent models of functional activity in the brain as a diffusion over the underlying anatomical pathways (Abdelnour et al., 2014; Liang & Wang, 2017), we show the applicability of GDNs to infer brain structural connectivity from functional networks obtained from the Human Connectome Project-Young Adult (HCP-YA) dataset. We also use GDNs to predict Facebook ties from user co-location data, outperforming relevant baselines.

**Related work.** Graph structure learning (a.k.a. network topology inference) has a long history in statistics (Dempster, 1972), with noteworthy contributions for probabilistic graphical model selection; see e.g. (Kolaczyk, 2009; Friedman et al., 2008; Drton & Maathuis, 2017). Recent advances were propelled by GSP insights through the lens of signal representation (Dong et al., 2019; Mateos et al., 2019; Giannakis et al., 2018), exploiting models of network diffusion (Daneshmand et al., 2014), or else leveraging cardinal properties of network data such as smoothness (Dong et al., 2016; Kalofolias, 2016) and graph stationarity (Segarra et al., 2017; Pasdeloup et al., 2018). These works formulate (convex) optimization problems one has to solve for different graphs, and can lack robustness to signal model misspecifications. Network deconvolution approaches in (Segarra et al., 2017; Feizi et al., 2013) operate in the graph spectral domain, and may face scalability issues because they rely on computationally-expensive eigendecompositions of the input graph for each problem instance. Moreover, none of these methods advocate a supervised learning paradigm using NNs to predict adjacency matrices as we propose here. In the broader context of NN-based models, so-termed latent graph learning has been shown effective in obtaining better task-driven representations of relational data for machine learning (ML) applications (Wang et al., 2019; Kazi et al., 2023; Veličković et al., 2020), or to learn interactions among coupled dynamical systems (Kipf et al., 2018). With regards to NN architectural design, algorithm unrolling has only recently been adopted for graph structure learning in (Shrivastava et al., 2020; Pu et al., 2021); but for different and more specific problems subsumed by the network deconvolution framework dealt with here. Indeed, (Shrivastava et al., 2020) focuses on Gaussian graphical model selection in a supervised learning setting and (Pu et al., 2021) learns graph topologies under a smoothness signal prior.

**Summary of main contributions.** In this work, we introduce the following graph ML innovations:

- We introduce GDNs, a supervised learning NN model capable of recovering sparse latent graph structure from observations of its convolutional mixtures, i.e., related graphs containing spurious, indirect relationships. The novel supervised setting for graph structure learning is motivated via several real-world applications.

- The node permutation equivariant GDN architecture is derived from the principle of algorithm unrolling to learn fast approximate solutions to inverse problems. The unrolling offers explicit control on complexity (leading to fast inference times) and can seamlessly integrate domain-specific prior information about the unknown graph distribution. In constructing GDNs we leverage Multi-Input Multi-Output (MIMO) filters – as used in convolutional (C)NNs – for further expressiveness and training stability.

- On synthetic data, GDNs outperform comparable methods on link prediction and edge-weight regression tasks across different random-graph ensembles, while incurring a markedly lower (post-training) computational cost and inference time. GDNs are inductive and learnt models transfer across graphs of different sizes. We empirically verify they exhibit minimal performance degradation even when tested on graphs $30\times$ larger.

• Finally, using GDNs we propose a novel ML pipeline to learn whole brain structural connectivity (SC) from functional connectivity (FC), a challenging and timely problem in network neuroscience that motivates the supervised learning setting advocated here. Results on the HCP-YA imaging dataset show that GDNs perform well on specific brain subnetworks that are known to be relatively less correlated with the corresponding FC due to ageing-related effects – a testament to the model's robustness and expressive power. Overall, results here support the promising prospect of using graph ML to integrate brain structure and function.

## 2   Problem Formulation

In this work we study the following network inverse problem involving undirected and weighted graphs $\mathcal{G}(\mathcal{V}, \mathcal{E})$, where $\mathcal{V} = \{1, \ldots, N\}$ is the set of nodes (henceforth common to all graphs), and $\mathcal{E} \subseteq \mathcal{V} \times \mathcal{V}$ collects the edges. We get to observe a graph with symmetric adjacency matrix $\boldsymbol{A}_O \in \mathbb{R}^{N \times N}$, that is related to a latent sparse, graph $\boldsymbol{A}_L \in \mathbb{R}_+^{N \times N}$ of interest via the forward data model

$$\boldsymbol{A}_O = h_0 \boldsymbol{I} + h_1 \boldsymbol{A}_L + \underbrace{h_2 \boldsymbol{A}_L^2 \ldots + h_K \boldsymbol{A}_L^K}_{\text{indirect relationships}} \tag{1}$$

for some $K \leq N - 1$ by the Cayley-Hamilton theorem. Matrix polynomials $\boldsymbol{H}(\boldsymbol{A}; \boldsymbol{h}) := \sum_{k=0}^{K} h_k \boldsymbol{A}^k$ with coefficients $\boldsymbol{h} := [h_0, \ldots, h_K]^\top \in \mathbb{R}^{K+1}$ as in the right-hand-side of eq. (1), are known as shift-invariant graph convolutional filters; see e.g., (Ortega et al., 2018; Gama et al., 2020b). We postulate that $\boldsymbol{A}_O = \boldsymbol{H}(\boldsymbol{A}_L; \boldsymbol{h})$ for some filter order $K$ and its associated coefficients $\boldsymbol{h}$, such that we can think of the observed network as generated via a graph convolutional process acting on $\boldsymbol{A}_L$. That is, one can think of $\boldsymbol{A}_O$ as a graph containing spurious, indirect connections generated by the terms including higher-order powers of latent graph $\boldsymbol{A}_L$ – the graph of fundamental relationships. More pragmatically $\boldsymbol{A}_O$ may correspond to a noisy observation of $\boldsymbol{H}(\boldsymbol{A}_L; \boldsymbol{h})$, and this will be clear from the context when e.g., we estimate $\boldsymbol{A}_O$ from data.

Recovery of the latent graph $\boldsymbol{A}_L$ is a challenging endeavour since we do not know $\boldsymbol{H}(\boldsymbol{A}_L; \boldsymbol{h})$, namely the parameters $K$ or $\boldsymbol{h}$; see Appendix A.1 for issues of model identifiability and their relevance to the problem dealt with here. Suppose that $\boldsymbol{A}_L$ is drawn from some distribution of sparse graphs, e.g., random geometric graphs or structural brain networks from a relatively homogeneous dataset. Then given independent training samples $\mathcal{T} := \{\boldsymbol{A}_O^{(i)}, \boldsymbol{A}_L^{(i)}\}_{i=1}^T$ adhering to eq. (1), our goal is to learn a parametric mapping $\Phi$ that predicts the graph adjacency matrix $\hat{\boldsymbol{A}}_L = \Phi(\boldsymbol{A}_O; \boldsymbol{\Theta})$ by minimizing a loss function

$$L(\boldsymbol{\Theta}) := \frac{1}{T} \sum_{i \in \mathcal{T}} \ell(\boldsymbol{A}_L^{(i)}, \Phi(\boldsymbol{A}_O^{(i)}; \boldsymbol{\Theta})) \tag{2}$$

to search for the best parameters $\boldsymbol{\Theta}$. The loss $\ell$ is adapted to the task at hand – hinge loss for link prediction or mean-squared error (MSE) for the more challenging edge-weight regression problem; see Appendix A.5. Notice that eq. (1) postulates a common filter across graph pairs in $\mathcal{T}$ – a simplifying assumption shown to be tenable in functional magnetic resonance imaging (fMRI) studies (Abdelnour et al., 2014) where subject-level coupling between the structural and functional modalities (SC-FC coupling) exhibit limited variability. This will be relaxed when we customize the GDN architecture to learn multiple filters; see Section 4.3 for details.

Potential application domains for which supervised network data $\mathcal{T} := \{\boldsymbol{A}_O^{(i)}, \boldsymbol{A}_L^{(i)}\}_{i=1}^T$ is available include bioinformatics (infer protein contact structure $\boldsymbol{A}_L$ from mutual information graphs $\boldsymbol{A}_O$ of the covariation of amino acid residues (Feizi et al., 2013)), gene regulatory network inference from microarray data (see e.g. the DREAM5 project to reconstruct networks for the E. coli bacterium and single-cell eukaryotes), social and information networks (e.g., learn to sparsify graphs (Spielman & Srivastava, 2011) to unveil the most relevant collaborations $\boldsymbol{A}_L$ in a social network encoding co-authorship information $\boldsymbol{A}_O$ (Segarra et al., 2017)), and epidemiology (such as contact tracing by deconvolving the graphs that model observed disease spread in a population). In Section 5.2 we experiment with social networks and the network neuroscience problem described next.

## 3 Motivating Application Domains and the Supervised Setting

Here we outline several graph structure learning tasks that can be cast as the network inverse problem in eq. (1), and give numerous examples supporting the relevance of the novel supervised setting.

**Graph structure identification from diffused signals.** Our initial focus is on identifying graphs that explain the structure of a class of network diffusion processes. Formally, let $\boldsymbol{x} \in \mathbb{R}^N$ be a graph signal (i.e., a vector of nodal features) supported on a latent graph $\mathcal{G}$ with adjacency $\boldsymbol{A}_L$. Further, let $\boldsymbol{w}$ be a zero-mean white signal with covariance matrix $\boldsymbol{\Sigma}_w = \mathbb{E}[\boldsymbol{w}\boldsymbol{w}^\top] = \boldsymbol{I}$. We say that $\boldsymbol{A}_L$ represents the structure of the signal $\boldsymbol{x}$ if there exists a linear network diffusion process in $\mathcal{G}$ that generates the signal $\boldsymbol{x}$ from $\boldsymbol{w}$, namely $\boldsymbol{x} = \sum_{i=0}^{\infty} h_i \boldsymbol{A}_L^i \boldsymbol{w} = \boldsymbol{H}(\boldsymbol{A}_L; \boldsymbol{h})\boldsymbol{w}$. This is a fairly common generative model for random network processes (Barrat et al., 2008; DeGroot, 1974). We think of the edges of $\mathcal{G}$ as direct (one-hop) relations between the elements of the signal $\boldsymbol{x}$. The diffusion described by $\boldsymbol{H}(\boldsymbol{A}_L; \boldsymbol{h})$ generates indirect relations. In this context, the graph structure learning problem is to recover a sparse $\boldsymbol{A}_L$ from a set $\mathcal{X} := \{\boldsymbol{x}_i\}_{i=1}^P$ of $P$ samples of $\boldsymbol{x}$ (Segarra et al., 2017). Interestingly, from the model for $\boldsymbol{x}$ it follows that the signal covariance matrix $\boldsymbol{\Sigma}_x = \mathbb{E}\left[\boldsymbol{x}\boldsymbol{x}^\top\right] = \boldsymbol{H}(\boldsymbol{A}_L; \boldsymbol{h})\mathbb{E}\left[\boldsymbol{w}\boldsymbol{w}^\top\right]\boldsymbol{H}(\boldsymbol{A}_L; \boldsymbol{h}) = \boldsymbol{H}^2(\boldsymbol{A}_L; \boldsymbol{h})$ is also a polynomial in $\boldsymbol{A}_L$ – precisely the relationship prescribed by eq. (1) when $\boldsymbol{A}_O = \boldsymbol{\Sigma}_x$. In practice, given the signals in $\mathcal{X}$ one would estimate the covariance matrix, e.g. via the sample covariance $\hat{\boldsymbol{\Sigma}}_x$, and then aim to recover the graph $\boldsymbol{A}_L$ by tackling the aforementioned network inverse problem. In this paper, we propose a fresh learning-based solution using training examples $\mathcal{T} := \{\hat{\boldsymbol{\Sigma}}_x^{(i)}, \boldsymbol{A}_L^{(i)}\}_{i=1}^T$.

**Remark** (Graph convolutional data model in context)**.** To further elaborate on the relevance and breadth of applicability of the graph convolutional (or network diffusion) signal model $\boldsymbol{x} = \boldsymbol{H}(\boldsymbol{A}_L; \boldsymbol{h})\boldsymbol{w}$, we would like to elucidate connections with related work for graph structure learning. Note that while we used the diffusion-based generative model to formulate the problem above, we do not need it as an actual mechanistic process. Indeed, like in eq. (1) the only thing we ask is for the data covariance $\boldsymbol{A}_O = \boldsymbol{\Sigma}_x$ to be some analytic function of the latent graph $\boldsymbol{A}_L$. This is not extraneous to workhorse statistical methods for network topology inference, which (implicitly) make specific choices for these mappings, e.g. (i) correlation networks (Kolaczyk, 2009, Ch. 7) rely on the identity mapping $\boldsymbol{\Sigma}_x = \boldsymbol{A}_L$; (ii) Gaussian graphical model selection methods, such as graphical lasso in (Yuan & Lin, 2007; Friedman et al., 2008), adopt $\boldsymbol{\Sigma}_x = \boldsymbol{A}_L^{-1}$; and (iii) undirected structural equation models $\boldsymbol{x} = \boldsymbol{A}_L\boldsymbol{x} + \boldsymbol{w}$ which implies $\boldsymbol{\Sigma}_x = (\boldsymbol{I} - \boldsymbol{A}_L)^{-2}$ (Mateos et al., 2019). Accordingly, these models are all subsumed by the general framework we put forth here. For a recent and inspiring supervised learning approach to Gaussian graphical model selection rooted on graphical lasso, the interested reader is referred to (Shrivastava et al., 2020).

**Network deconvolution and denoising.** The network deconvolution problem is to identify a sparse adjacency matrix $\boldsymbol{A}_L$ that encodes direct dependencies, when given an adjacency matrix $\boldsymbol{A}_O$ containing extraneous indirect relationships. The problem broadens the scope of signal deconvolution to networks and can be tackled by attempting to invert the mapping $\boldsymbol{A}_O = \boldsymbol{A}_L(\boldsymbol{I} - \boldsymbol{A}_L)^{-1} = \boldsymbol{A}_L + \boldsymbol{A}_L^2 + \boldsymbol{A}_L^3 \dots$ This solution proposed in (Feizi et al., 2013) assumes a polynomial relationship as in eq. (1), but for the particular case of a single-pole, single-zero graph filter with very specific filter coefficients [cf. eq. (1) with $h_0 = 0$ and $h_k = 1$, $k \geq 1$]. This way, the indirect dependencies observed in $\boldsymbol{A}_O$ arise due to the higher-order convolutive mixture terms $\boldsymbol{A}_L^2 + \boldsymbol{A}_L^3 + \dots$ superimposed to the direct interactions in $\boldsymbol{A}_L$ we wish to recover. Here we adopt a general data-driven learning approach in assuming that $\boldsymbol{A}_O$ can be written as a polynomial in $\boldsymbol{A}_L$, but being agnostic to the form of the filter, thus broadening the problem setting in (Feizi et al., 2013). Unlike the problem outlined in the previous subsection, here $\boldsymbol{A}_O$ need not be a covariance matrix. Indeed, $\boldsymbol{A}_O$ could be a corrupted graph we wish to denoise, obtained via an upstream graph learning method. A related but different deconvolution problem was put forth in (Li et al., 2021). Rather than recovering graph structure, the goal therein is to reconstruct the input graph signals from smoothed nodal representations generated by a graph convolutional network (GCN) (Kipf & Welling, 2017).

**Inferring structural brain networks from functional MRI (fMRI) signals.** Brain connectomes encompass networks of brain regions connected by (statistical) functional associations (FC) or by anatomical white matter fiber pathways (SC). The latter can be extracted from time-consuming tractography algorithms applied to diffusion MRI (dMRI), which are particularly fraught due to quality issues in the data (Yeh et al., 2021). FC represents pairwise correlation structure between blood-oxygen-level-dependent (BOLD) signals

measured by fMRI. Deciphering the relationship between SC and FC is a very active area of research (Abdelnour et al., 2014; Honey et al., 2009) and also relevant in studying neurological disorders, since it is known to vary with respect to healthy subjects in pathological contexts (Gu et al., 2021). Traditional approaches exploring the SC-FC coupling go all the way from correlation studies (Greicius et al., 2008) to large-scale simulations of nonlinear cortical activity models (Honey et al., 2009). More aligned with the problem addressed here, recent studies have shown that linear diffusion dynamics can reasonably model the SC-FC coupling (Abdelnour et al., 2014). Using our notation, the findings in (Abdelnour et al., 2014) suggest that the covariance $\boldsymbol{A}_O = \boldsymbol{\Sigma}_x$ of the functional signals (i.e., the FC graph) is related to the the sparse SC network $\boldsymbol{A}_L$ via the model in eq. (1). Similarly, (Liang & Wang, 2017) contend *FC can be modeled as a weighted sum of powers of the SC matrix*, consisting of both direct and indirect effects along varying paths. There is evidence that FC links tend to exist where there is no or little structural connection (Damoiseaux & Greicius, 2009), a property naturally captured by eq. (1). These considerations motivate adopting our graph strcuture learning method to infer SC patterns from fMRI signals using the training set $\mathcal{T} := \{\mathbf{FC}^{(i)}, \mathbf{SC}^{(i)}\}_{i=1}^T$ (Section 5.2), a significant problem for several reasons. The ability to collect only FC and get informative estimates of SC open the door to large-scale studies, previously constrained by the logistical, cost, and computational resources needed to acquire both modalities.

## 4 Graph Deconvolution Network

Here we present the proposed GDN model, a NN architecture for graph structure learning that we train in a supervised fashion. In the sequel, we obtain 'conceptual' iterations to tackle an optimization formulation of the network inverse problem (Section 4.1), unroll these iterations to arrive at the parametric, differentiable GDN function $\Phi(\boldsymbol{A}_O; \boldsymbol{\Theta})$ we train using graph data $\mathcal{T}$ (Section 4.2), and describe architectural customizations to improve performance (Section 4.3).

### 4.1 Iterative optimization as NN architectural blueprint

Going back to the inverse problem of recovering a sparse adjacency matrix $\boldsymbol{A}_L$ from the mixture $\boldsymbol{A}_O$ in eq. (1), if the graph filter $\boldsymbol{H}(\boldsymbol{A}; \boldsymbol{h})$ were known – but recall it is not – we could attempt to solve

$$\hat{\boldsymbol{A}}_L \in \underset{\boldsymbol{A} \in \mathcal{A}}{\arg\min} \left\{ \|\boldsymbol{A}\|_1 + \frac{\lambda}{2} \|\boldsymbol{A}_O - \boldsymbol{H}(\boldsymbol{A}; \boldsymbol{h})\|_F^2 \right\}, \tag{3}$$

where $\lambda > 0$ trades off sparsity for reconstruction error. The convex set $\mathcal{A} := \{\boldsymbol{A} \in \mathbb{R}^{N \times N} \mid \mathrm{diag}(\boldsymbol{A}) = \boldsymbol{0}, A_{ij} = A_{ij} \geq 0, \forall i, j \in \{1, \ldots, N\}\}$ encodes the admissibility constraints on the adjacency matrix of an undirected graph: hollow diagonal, symmetric, with non-negative edge weights. The $\ell_1$ norm encourages sparsity in the solution, being a convex surrogate of the edge-cardinality function that counts the number of non-zero entries in $\boldsymbol{A}$ (Tibshirani, 1996). Since $\boldsymbol{A}_O$ is often a noisy observation or estimate of the polynomial $\boldsymbol{H}(\boldsymbol{A}_L; \boldsymbol{h})$, it is prudent to relax the equality in eq. (1) and minimize the squared residual errors instead. As described earlier, this could be the case when $\boldsymbol{A}_O = \hat{\boldsymbol{\Sigma}}_x$ (e.g., brain FC) and the empirical covariance matrices are estimated from finite data.

The composite cost in problem (3) is a weighted sum of a non-smooth function $\|\boldsymbol{A}\|_1$ and a continuously differentiable, non-convex function $g(\boldsymbol{A}) := \frac{1}{2} \|\boldsymbol{A}_O - \boldsymbol{H}(\boldsymbol{A}; \boldsymbol{h})\|_F^2$. But our end goal here is not to solve problem (3) iteratively, recall we cannot even formulate the optimization problem because $\boldsymbol{H}(\boldsymbol{A}; \boldsymbol{h})$ is unknown. To retain the essence of the problem structure and motivate a parametric model to learn approximate solutions from $\mathcal{T}$, it suffices to settle with 'conceptual' proximal gradient (PG) iterations ($k$ henceforth denote iterations, $\boldsymbol{A}[0] \in \mathcal{A}$)

$$\boldsymbol{A}[k+1] = \mathrm{ReLU}\left(\boldsymbol{A}[k] - \tau \nabla g(\boldsymbol{A}[k]) - \tau \boldsymbol{1}\boldsymbol{1}^\top\right) \tag{4}$$

for $k = 0, 1, 2, \ldots$, where $\tau$ is a step-size parameter in which we have absorbed $\lambda$. These iterations implement a gradient descent step on $g$ followed by the $\ell_1$ norm's proximal operator; see Appendix A.2 for a detailed derivation of eq. (4) and (Parikh & Boyd, 2014) for more on PG algorithms. Due to the non-negativity constraints in $\mathcal{A}$, the $\ell_1$ norm's proximal operator takes the form of a $\tau$-shifted ReLU on the off-diagonal entries of its matrix argument. Also, the operator sets $\mathrm{diag}(\boldsymbol{A}[k+1]) = \boldsymbol{0}$.

We cannot run the iterates in (4) without knowledge of $\boldsymbol{h}$, and we will make no attempt to estimate $\boldsymbol{h}$. Instead, our approach in the next section is to unroll and truncate these iterations (and in the process, convert unknowns to learnable parameters $\boldsymbol{\Theta}$) to arrive at the trainable GDN parametric model $\Phi(\boldsymbol{A}_O; \boldsymbol{\Theta})$. This way, the iterations in (4) will serve as NN architectural blueprint to tackle the supervised problem stated in Section 2.

### 4.2 Learning to infer graphs via algorithm unrolling

The idea of algorithm unrolling can be traced back to the seminal work of (Gregor & LeCun, 2010). In the context of sparse coding, they advocated identifying *iterations* of PG algorithms with *layers* in a deep network of fixed depth that can be trained from examples using backpropagation. One can view this process as effectively truncating the iterations of an asymptotically convergent procedure, to yield a template architecture that learns to approximate solutions with substantial computational savings relative to the optimization algorithm. Beyond parsimonious signal modeling, there has been a surge in popularity of unrolled deep networks for a wide variety of applications; see e.g., (Monga et al., 2021) for a recent tutorial treatment focused on signal and image processing.

However, to the best of our knowledge this approach is yet to be fully explored for graph structure learning. L2G (Pu et al., 2021) and GLAD (Shrivastava et al., 2020) represent recent inspiring attempts, but neither are NNs (meaning compositions of affine layers followed by point-wise nonlinear activations). Both process the input data and the hidden representations in a nonlinear fashion (pre activation) involving, e.g., cross-parameter products or parameter inverses within their layers, challenging optimization during training (Monga et al., 2021). GLAD includes a matrix square root operation which poses scalability issues due to its cubic complexity. A detailed account of the differences between GDN and GLAD is given in Appendix A.7.

Building on the algorithm unrolling paradigm and starting from the PG iterations (4), we design a non-linear, parameterized, feed-forward NN architecture that can be trained to predict the latent graph $\hat{\boldsymbol{A}}_L = \Phi(\boldsymbol{A}_O; \boldsymbol{\Theta})$. We aim at a NN architecture for its favorable optimization properties, yet desire to retain expressiveness. To this end, we approximate the gradient $\nabla g(\boldsymbol{A})$ by retaining only linear terms in $\boldsymbol{A}$, and build a deep network by composing layer-wise linear filters and point-wise nonlinearites to capture higher-order interactions in the generative process $\boldsymbol{H}(\boldsymbol{A}; \boldsymbol{h}) := \sum_{k=0}^{K} h_k \boldsymbol{A}^k$. This rationale is similar to the one followed in developing the GCN model (Kipf & Welling, 2017). Therein, it is argued that a rich class of convolutional filter functions (e.g., those parameterized via Chebyshev polynomials) can be recovered by stacking multiple linear GCN layers.

In more detail, we start by simplifying $\nabla g(\boldsymbol{A})$ (derived in Appendix A.2) by dropping all higher-order terms in $\boldsymbol{A}$, namely

$$\nabla g(\boldsymbol{A}) = -\sum_{k=1}^{K} h_k \sum_{r=0}^{k-1} \boldsymbol{A}^{k-r-1} \boldsymbol{A}_O \boldsymbol{A}^r + \frac{1}{2} \nabla_{\boldsymbol{A}} \operatorname{Tr}\left[\boldsymbol{H}^2(\boldsymbol{A}; \boldsymbol{h})\right]$$
$$\approx -h_1 \boldsymbol{A}_O - h_2(\boldsymbol{A}_O \boldsymbol{A} + \boldsymbol{A} \boldsymbol{A}_O) + (2h_0 h_2 + h_1^2)\boldsymbol{A}. \tag{5}$$

Notice that $\nabla_{\boldsymbol{A}} \operatorname{Tr}\left[\boldsymbol{H}^2(\boldsymbol{A}; \boldsymbol{h})\right]$ is a polynomial of degree $2K-1$. Hence, we keep the linear term in $\boldsymbol{A}$ but drop the constant offset that is proportional to the identity matrix $\boldsymbol{I}$, which is inconsequential to adjacency matrix updates with null diagonal. An affine approximation leads to more benign optimization landscapes when it comes to training the resulting GDN model; see Section 5 for ablation studies exploring the effects of higher-order terms in the approximation. All in all, the simplified PG iterations become

$$\boldsymbol{A}[k+1] = \operatorname{ReLU}\left(\alpha \boldsymbol{A}[k] + \beta(\boldsymbol{A}_O \boldsymbol{A}[k] + \boldsymbol{A}[k]\boldsymbol{A}_O) + \gamma \boldsymbol{A}_O - \tau \mathbf{1}\mathbf{1}^\top\right), \tag{6}$$

where $\boldsymbol{A}[0] \in \mathcal{A}$ and we defined $\alpha := (1 - 2\tau h_0 h_2 - \tau h_1^2)$, $\beta := \tau h_2$, and $\gamma := \tau h_1$. The latter parameter triplet encapsulates filter (i.e., mixture) coefficients and the $\lambda$-dependent algorithm step-size $\tau$, all of which are unknown in practice.

The GDN architecture is thus obtained by unrolling the iterations (6) into a deep NN; see Figure 1. This entails mapping each individual iteration into a layer and stacking a prescribed number $D$ of layers together

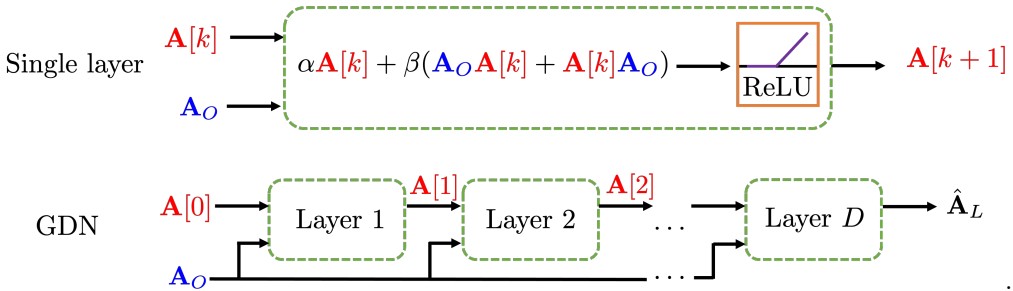

Figure 1: Schematic diagram of the GDN architecture obtained via algorithm unrolling.

to form $\Phi(\boldsymbol{A}_O; \boldsymbol{\Theta})$. The unknown filter coefficients are treated as learnable parameters $\boldsymbol{\Theta} := \{\alpha, \beta, \gamma, \tau\}$, which are shared across layers as in recurrent neural networks (RNNs). The reduced number of parameters relative to most typical NNs is a characteristic of unrolled networks (Monga et al., 2021). A first noteworthy property of the GDN layer in eq. (6) is that it preserves symmetry, namely if the input $\boldsymbol{A}[k]$ to layer $k+1$ is a symmetric matrix then so is the output $\boldsymbol{A}[k+1]$. Second, GDN parametrizations are equivariant to node permutations – an essential property that guarantees the learnt graph topologies are not affected by the order in which we label the vertices.

**Proposition 1** (Permutation equivariance of GDNs). Let $\boldsymbol{P} \in \{0,1\}^{N \times N}$ be a permutation matrix and consider identical row and column permutations of the observed graph $\tilde{\boldsymbol{A}}_O = \boldsymbol{P}^\top \boldsymbol{A}_O \boldsymbol{P}$ as well as the initial state $\tilde{\boldsymbol{A}}[0] = \boldsymbol{P}^\top \boldsymbol{A}[0] \boldsymbol{P}$. Let $\Phi(\boldsymbol{A}_O; \boldsymbol{\Theta}, \boldsymbol{A}[0])$ denote the GDN output corresponding to input $\boldsymbol{A}_O$, where for convenience we make explicit that the model is parameterized by the initial state $\boldsymbol{A}[0]$. Then, it holds that GDNs are permutation equivariant, namely

$$\Phi(\tilde{\boldsymbol{A}}_O; \boldsymbol{\Theta}, \tilde{\boldsymbol{A}}[0]) = \boldsymbol{P}^\top \Phi(\boldsymbol{A}_O; \boldsymbol{\Theta}, \boldsymbol{A}[0]) \boldsymbol{P}. \tag{7}$$

*Proof.* Let $\boldsymbol{A}[k+1] = f(\boldsymbol{A}[k], \boldsymbol{A}_O)$ represent the layer $k+1$ mapping in eq. (6). Then by regrouping terms, recalling that $\boldsymbol{P}^\top \boldsymbol{P} = \boldsymbol{I}$, and using the fact that ReLU($\cdot$) and $\mathbf{11}^\top$ are both permutation equivariant, we find via simple algebraic manipulations that $f(\boldsymbol{P}^\top \boldsymbol{A}[k] \boldsymbol{P}, \tilde{\boldsymbol{A}}_O) = \boldsymbol{P}^\top f(\boldsymbol{A}[k], \boldsymbol{A}_O) \boldsymbol{P}$. Because each layer satisfies the permutation equivariance property, the GDN architecture $\Phi(\boldsymbol{A}_O; \boldsymbol{\Theta}, \boldsymbol{A}[0])$ that is a composition of $D$ such layers is also permutation equivariant; meaning that eq. (7) holds true. □

In the next section, we will explore a few customizations to the vanilla GDN architecture in order to broaden the model's expressive power. Given a training set $\mathcal{T} = \{\boldsymbol{A}_O^{(i)}, \boldsymbol{A}_L^{(i)}\}_{i=1}^T$, learning is accomplished by using mini-batch stochastic gradient descent to minimize the task-dependent loss function $L(\boldsymbol{\Theta})$ in eq. (2). We adopt a hinge loss for link prediction and mean-squared/absolute error for the edge-weight regression task. For link prediction, we also learn a threshold $t \in \mathbb{R}_+$ to binarize the estimated edge weights and declare presence or absence of edges; see Appendix A.5 for all training-related details.

The iterative refinement principle of optimization algorithms carries over to our GDN model during inference. Indeed, we start with an initial estimate $\boldsymbol{A}[0] \in \mathcal{A}$ and use a cascade of $D$ linear filters and point-wise nonlinearities to refine it to an output $\hat{\boldsymbol{A}}_L = \Phi(\boldsymbol{A}_O; \hat{\boldsymbol{\Theta}})$; see Fig. 2 for an example drawn from the experiments in Section 5.1. Matrix $\boldsymbol{A}[0]$ is a hyperparameter we can select to incorporate prior information on the sought latent graph, or it could be learned as it is customary with the initial state in RNNs; see Section 4.3. The input graph $\boldsymbol{A}_O$ which we aim to deconvolve is directly fed to all layers, and contributes to defining non-uniform soft thresholds $\gamma \boldsymbol{A}_O - \tau \mathbf{11}^\top$ which sparsify the layer's output. One can also interpret $\alpha \boldsymbol{A} + \beta(\boldsymbol{A}_O \boldsymbol{A} + \boldsymbol{A} \boldsymbol{A}_O)$ as a first-order, symmetry-preserving filter defined on graph $\boldsymbol{A}_O$, which is used to process $\boldsymbol{A} \in \mathcal{A}$ – here viewed as a structured graph signal with $N$ features per node to invoke this GSP insight. In its simplest rendition, the GDN leverages elements of RNNs and GCNs (Kipf & Welling, 2017).

## 4.3 GDN architecture customizations

Here we outline several customizations and enhancements to the vanilla GDN architecture of the previous section, which we have empirically found to improve graph structure learning performance.

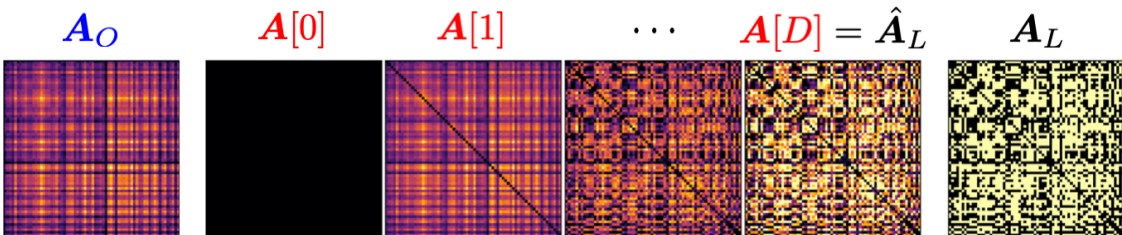

Figure 2: Intermediate outputs of a GDN model trained on the random geometric (RG) graphs in Table 1. GDNs refine the input $\boldsymbol{A}_O$ to more closely resemble $\boldsymbol{A}_L$ as we go deeper in the network. Notice how the predicted output adjacency matrix $\hat{\boldsymbol{A}}_L$ recovers the connectivity structure in $\boldsymbol{A}_L$.

**Incorporating prior information via algorithm initialization.** By viewing our method as an iterative refinement of an initial graph $\boldsymbol{A}[0]$, one can think of $\boldsymbol{A}[0]$ as a best initial guess, or *prior*, over $\boldsymbol{A}_L$. A simple strategy to incorporate prior information about some edge $(i, j)$, encoded in $A_{ij}$ that we view as a random variable, would be to set $A[0]_{ij} = \mathbb{E}[A_{ij}]$. This technique is adopted when training on the HCP-YA dataset in Section 5, by taking the prior $\boldsymbol{A}[0]$ to be the sample mean of all latent (i.e., SC) graphs in the training set $\mathcal{T}$. This encodes our prior knowledge that there are strong similarities in the structure of the human brain across the population of healthy young adults. When $\boldsymbol{A}_L$ is expected to be reasonably sparse, we can set $\boldsymbol{A}[0] = \boldsymbol{0}$, which is effective as we show in Table 1. Recalling the connections drawn between GDNs and RNNs, then the prior $\boldsymbol{A}[0]$ plays a similar role to the initial RNN state and thus it could be learned (Hinton, 2013). In any case, the ability to seamlessly incorporate prior information to the model through the initial state is an attractive feature of unrolling-based NNs (Monga et al., 2021) and of GDNs in particular.

**Multi-Input Multi-Output (MIMO) filters.** So far, in each layer we have a single learned filter, which takes an $N \times N$ matrix as input and returns another $N \times N$ matrix at the output. After going through the shifted ReLU nonlinearity, this refined output adjacency matrix is fed as input to the next layer; a process that repeats $D$ times. More generally, we can allow for multiple input channels (i.e., a tensor), as well as multiple channels at the output, by using the familiar convolutional neural network (CNN) methodology. This way, each output channel has its own filter parameters associated with every input channel. The $j$-th output channel applies its linear filters to all input channels, aggregating the results with a reduction operation (e.g., mean or sum), and applies a point-wise nonlinearity (here a shifted ReLU) to the output. This allows the GDN model to learn many different filters, thus providing richer learned representations and effectively relaxing the simplifying assumption we made regarding a common filter across graph pairs in eq. (1). Notice that the aforementioned enhancement to the GDN architecture does not compromise its permutation equivariance property. Full details of MIMO filters are in Appendix A.4.

**Decoupling layer parameters.** Thus far, we have respected the parameter sharing constraint imposed by the unrolled PG iterations. We now allow each layer to learn a decoupled MIMO filter, with its own set of parameters mapping from $C_{in}^k$ input channels to $C_{out}^k$ output channels. As the notation suggests, $C_{in}^k$ and $C_{out}^k$ need not be equal. By decoupling the layer structure, we allow GDNs to compose different learned filters to create more abstract features (as with CNNs or GCNs). Accordingly, it opens up the architectural design space to broader exploration, e.g., wider layers early and skinnier layers at the end. Exploring this architectural space is beyond the scope of this paper and is left as future work. In experiments, we use GDNs for which intermediate layers $k \in \{2, \dots, D-1\}$ have $C = C_{in}^k = C_{out}^k$, i.e., a flat architecture. We denote models with layer-wise shared parameters and independent parameters as GDN-S and GDN, respectively.

## 5 Experiments

We present comprehensive experiments on link prediction and edge-weight regression tasks using synthetic data (Section 5.1) as well as real HCP-YA neuroimaging and social network data (Section 5.2).

**Graph generation.** In the synthetic data experiments we consider three test cases whereby the latent graphs are respectively drawn from ensembles of Erdős-Rényi (ER), random geometric (RG), and Barabási-Albert

(BA) random graph models; see e.g., (Kolaczyk, 2009). We study an additional scenario where we use SCs from HCP-YA (referred to as the 'pseudo-synthetic' case because the latent graphs are real structural brain networks, while the signals used to estimate covariance matrices $\boldsymbol{A}_O = \hat{\boldsymbol{\Sigma}}_x$ are synthetic; see Section 5.1).

**Figures of merit.** In the link prediction task, performance is evaluated using error $:= \frac{\text{incorrectly predicted edges}}{\text{total possible edges}}$. For regression, we adopt the mean-squared-error (MSE) or mean-absolute-error (MAE) as figures of merit.

**Baselines.** We compare GDNs against several relevant graph structure learning baselines: Network Deconvolution (ND) which uses a spectral approach to directly invert a very specific convolutive mixture (Feizi et al., 2013); Spectral Templates (SpecTemp) that advocates a convex optimization approach to recover sparse graphs from noisy estimates of $\boldsymbol{A}_O$'s eigenvectors (Segarra et al., 2017); Graphical LASSO (GLASSO), a regularized MLE of the precision matrix for Gaussian graphical model selection (Friedman et al., 2008); a learned unrolling of alternating minimization (AM) on the GLASSO objective (GLAD) (Shrivastava et al., 2020); a learned unrolling of a primal-dual splitting (PDS) algorithm to minimize a graph recovery criterion which promotes a signal smoothness prior (L2G) (Pu et al., 2021); a GNN encoder-decoder model on graph $\boldsymbol{A}_O$ using the degree vector $\boldsymbol{A}_O\mathbf{1}$ as nodal signals, with encoders based on the GCN (GCN-D) or the graph isomorphism network (Xu et al., 2019) (GIN-D); least-squares fitting of $\boldsymbol{h}$ followed by non-convex optimization to solve problem (3) (LSOpt); and Hard Thresholding (Threshold) to assess how well a simple cutoff rule can perform. GLASSO, ND, and SpecTemp generally tune their regularization parameters in an unsupervised way, but here they are tuned with supervision; see A.7 for details on all baselines we implemented. The network inverse problem only assumes knowledge of the observed graph $\boldsymbol{A}_O$ for inference - *there is no nodal information available and GDN layers do not operate on node features*; see Appendix A.4 for a discussion on how GDNs can be extended to handle node features. As such, popular GNN-based methods used in the link-prediction task (Wang et al., 2019; Kazi et al., 2023; Veličković et al., 2020; Kipf et al., 2018; Kipf & Welling, 2016; Zhang & Chen, 2018) are not directly applicable here. We carefully reviewed the network inference literature and chose the most performant baselines available for this problem setting.

**GDN-related hyperparameters and experimental setting.** Unless otherwise stated, in all the results that follow we use GDN(-S) models with $D = 8$ layers, $C = 8$ channels per layer, take prior $\boldsymbol{A}[0] = \mathbf{0}$ on all domains except the SCs - where we use the sample mean of all SCs in the training set, and train using the ADAM optimizer (Kingma & Ba, 2015) with learning rate of 0.01 and batch size of 200. We use one Nvidia T4 GPU; models take $< 15$ minutes to train on all datasets. A detailed account on the GDN hyperparameter search process is included in the supplementary material; see Appendix A.8.

## 5.1 Learning graph structure from diffused signals

A set of latent graphs are either sampled from RG, ER, or the BA model, or taken as the SCs from the HCP-YA dataset. In an attempt to make the synthetic latent graphs somewhat comparable to the 68 node SCs, we sample connected $N = 68$ node graphs with edge densities in the range $[0.5, 0.6]$ when feasible – typical values observed in SC graphs. To generate each observation $\boldsymbol{A}_O^{(i)}$, we simulated $P = 50$ standard Normal white signals diffused over $\boldsymbol{A}_L^{(i)}$; from which we form the sample covariance $\hat{\boldsymbol{\Sigma}}_x$ as described in Section 3. The value of $P$ is chosen based on a study of robustness to noise in the observations across multiple models, which showed only marginally improved performance when averaging more signals as shown in Table 2. With respect to the diffusion graph filters we let $K = 2$, and sample the filter coefficients $\boldsymbol{h} \in \mathbb{R}^3$ in $\boldsymbol{H}(\boldsymbol{A}_L; \boldsymbol{h})$ uniformly from the unit sphere. To examine robustness to the realizations of $\boldsymbol{h}$, we repeat this data generation process three times (resampling the filter coefficients). We thus create three different datasets for each graph domain (12 in total). For the sizes of the training/validation/test splits, the pseudo-synthetic domain uses 913/50/100 and the synthetic domains use 913/500/500.

Table 1 tabulates the results for synthetic and pseudo-synthetic experiments. For graph models that exhibit localized connectivity patterns (RG and SC), GDNs significantly outperform the baselines on both tasks. For the SC test case, GDN (GDN-S) reduces error relative to the mean prior by $27.5 \pm 1.7\%$ ($23.0 \pm 1.7\%$) and MSE by $37.3 \pm 0.8\%$ ($23.2 \pm 0.5\%$). Both GDN architectures show the ability to learn such local patterns, with the extra representational power of GDNs (over GDN-S) providing an additional boost in performance. All models struggle on BA and ER with GDNs showing better performance even for these cases.

Table 1: Mean and standard error of the test performance across both tasks (Top: link-prediction, Bottom: edge-weight regression) on each graph domain. Bold denotes best performance.

| | Models | RG | ER | BA | SC |
|---|---|---|---|---|---|
| **Error (%)** | GDN | $\mathbf{4.6}_{\pm\text{4e-1}}$ | $41.9_{\pm\text{1e-1}}$ | $\mathbf{27.5}_{\pm\text{1e-3}}$ | $\mathbf{8.9}_{\pm\text{2e-2}}$ |
| | GDN-S | $5.5_{\pm\text{2e-1}}$ | $\mathbf{40.8}_{\pm\text{1e-2}}$ | $27.6_{\pm\text{8e-4}}$ | $9.4_{\pm\text{2e-1}}$ |
| | L2G | $44.3_{\pm\text{3e-4}}$ | $43.9_{\pm\text{2e-5}}$ | $34.9_{\pm\text{1e-5}}$ | $31.7_{\pm\text{3e-1}}$ |
| | GCN-D | $30.7_{\pm\text{3e-4}}$ | $41.8_{\pm\text{1e-5}}$ | $27.6_{\pm\text{5e-5}}$ | $31.6_{\pm\text{9e-5}}$ |
| | GIN-D | $20.0_{\pm\text{2e-3}}$ | $43.6_{\pm\text{2e-3}}$ | $28.3_{\pm\text{8e-4}}$ | $33.3_{\pm\text{8e-3}}$ |
| | GLAD | $6.3_{\pm\text{2e-1}}$ | $43.7_{\pm\text{3e-1}}$ | $35.0_{\pm\text{3e-2}}$ | $11.8_{\pm\text{2e-3}}$ |
| | GLASSO | $8.8_{\pm\text{7e-2}}$ | $43.2_{\pm\text{1e-2}}$ | $34.9_{\pm\text{9e-3}}$ | $20.0_{\pm\text{4e-2}}$ |
| | ND | $9.4_{\pm\text{3e-1}}$ | $43.9_{\pm\text{1e-2}}$ | $34.1_{\pm\text{8e-3}}$ | $21.3_{\pm\text{9e-2}}$ |
| | SpecTemp | $11.1_{\pm\text{3e-1}}$ | $44.4_{\pm\text{7e-2}}$ | $30.2_{\pm\text{2e-1}}$ | $30.0_{\pm\text{1e-1}}$ |
| | LSOpt | $24.2_{\pm\text{5e-0}}$ | $42.5_{\pm\text{3e-1}}$ | $28.0_{\pm\text{2e-1}}$ | $31.5_{\pm\text{6e-3}}$ |
| | Threshold | $26.8_{\pm\text{2e-1}}$ | $42.9_{\pm\text{8e-1}}$ | $32.3_{\pm\text{1e-0}}$ | $21.7_{\pm\text{2e-1}}$ |
| **MSE** | GDN | $\mathbf{4.2}\text{e-}\mathbf{2}_{\pm\text{4e-3}}$ | $2.3\text{e-}1_{\pm\text{2e-3}}$ | $\mathbf{1.8}\text{e-}\mathbf{1}_{\pm\text{2e-3}}$ | $\mathbf{5.3}\text{e-}\mathbf{3}_{\pm\text{7e-5}}$ |
| | GDN-S | $6.0\text{e-}2_{\pm\text{2e-1}}$ | $\mathbf{2.3}\text{e-}\mathbf{1}_{\pm\text{2e-3}}$ | $2.7\text{e-}1_{\pm\text{2e-2}}$ | $6.5\text{e-}3_{\pm\text{4e-5}}$ |
| | L2G | $2.5\text{e-}1_{\pm\text{4e-4}}$ | $2.5\text{e-}1_{\pm\text{3e-5}}$ | $2.3\text{e-}1_{\pm\text{4e-5}}$ | $4.9\text{e-}2_{\pm\text{3e-3}}$ |
| | GCN-D | $2.1\text{e-}1_{\pm\text{5e-6}}$ | $2.4\text{e-}1_{\pm\text{3e-5}}$ | $2.0\text{e-}1_{\pm\text{1e-5}}$ | $1.1\text{e-}1_{\pm\text{6e-5}}$ |
| | GIN-D | $2.0\text{e-}1_{\pm\text{3e-3}}$ | $2.6\text{e-}1_{\pm\text{5e-3}}$ | $1.9\text{e-}1_{\pm\text{7e-4}}$ | $2.6\text{e-}1_{\pm\text{5e-3}}$ |
| | GLAD | $7.8\text{e-}2_{\pm\text{9e-4}}$ | $2.4\text{e-}1_{\pm\text{4e-3}}$ | $1.9\text{e-}1_{\pm\text{4e-3}}$ | $1.4\text{e-}2_{\pm\text{7e-6}}$ |
| | GLASSO | $2.0\text{e-}1_{\pm\text{3e-3}}$ | $2.8\text{e-}1_{\pm\text{2e-2}}$ | $2.6\text{e-}1_{\pm\text{2e-2}}$ | $4.4\text{e-}2_{\pm\text{3e-5}}$ |
| | ND | $1.8\text{e-}1_{\pm\text{2e-3}}$ | $2.4\text{e-}1_{\pm\text{5e-4}}$ | $2.2\text{e-}1_{\pm\text{1e-3}}$ | $5.6\text{e-}2_{\pm\text{7e-5}}$ |
| | SpecTemp | $5.1\text{e-}2_{\pm\text{3e-5}}$ | $5.3\text{e-}1_{\pm\text{9e-5}}$ | $3.3\text{e-}1_{\pm\text{2e-5}}$ | $1.5\text{e-}1_{\pm\text{4e-3}}$ |
| | LSOpt | $9.9\text{e-}2_{\pm\text{2e-1}}$ | $2.5\text{e-}1_{\pm\text{2e-3}}$ | $2.0\text{e-}1_{\pm\text{3e-3}}$ | $6.1\text{e-}0_{\pm\text{6e-4}}$ |

**Robustness to noise.** To evaluate the robustness of GDN's to noise in the observations $\boldsymbol{A}_O$, we train a GDN model with $D = 30$ and $C = 1$ on multiple datasets with varying levels of noise. We impliclty control the level of noise by changing the number of signals $P$ used to estimate $\boldsymbol{A}_O$ - taken to be the sample covariance matrix $\hat{\boldsymbol{\Sigma}}_x$. The data are the same as in Table 1, except here we perform diffusions using one of the sampled filter coefficients $\boldsymbol{h} \in \mathbb{R}^3$. The results of this experiment on both GDN and Threshold are reported in Table 2. We find that GDN's are robust to noisy observed graphs $\boldsymbol{A}_O$, achieving less than 10% error with only 20 observed signals. Up to $P = 50$, GDN performs significantly better when given more signals, with only marginal increases in performance after this point. This experiment also highlights the advantage of MIMO filters as a non-MIMO GDN of depth 30 is outperformed by the MIMO GDN of depth 8 on the $P = 50$ case.

**Scaling and size generalization: Deploying on larger graphs.** GDNs learn the parameters of graph convolutions for the processing of graphs making them inductive: we can deploy the learnt model on larger graph size domains. Such a deployment is feasible on larger graphs due to the $\mathcal{O}(N^2)$ time and memory complexity of GDNs; the same is true for L2G. This stands in contrast to the $\mathcal{O}(N^3)$ time complexity of SpecTemp, GLASSO, ND, and GLAD. SpecTemp and ND perform a full eigendecomposition of $\boldsymbol{A}_O$, the iterative algorithms for solving GLASSO incur cubic worst-case complexity (Zhang et al., 2018), and GLAD performs a matrix square root in *each* layer. GLAD also requires deeper unrolled networks, discounted intermediate losses, and multiple embedded MLPs, which have significant practical implications on memory usage; see also Appendix A.7. To substantiate this qualitative discussion, in Appendix A.8 we have included supplementary experimental runtime and memory usage analyses showing GDN compares favorably to SpecTemp, L2G, and GLAD.

To test the extent to which GDNs generalize when $N$ grows, we trained GDN(-S) on RG graphs with size $N = 68$, and tested them on RG graphs of size $N = [68, 75, 100, 200, 500, 1000, 2000]$, with 200 graphs of each size. As graph sizes increase, we require more samples in the estimation of the sample covariance to maintain a constant signal-to-noise ratio. To simplify the experiment and its interpretation, we disregard estimation

Table 2: Mean and standard error of the test performance on link prediction task for a varied number $P$ of sampled signals on $N = 68$ RG graphs.

| Models  \  P | 3 | 5 | 20 | 50 | 100 | 200 | 1000 |
|---|---|---|---|---|---|---|---|
| GDN | $26.4_{\pm 2e\text{-}3}$ | $17.9_{\pm 2e\text{-}3}$ | $9.6_{\pm 9e\text{-}4}$ | $7.3_{\pm 8e\text{-}4}$ | $6.8_{\pm 8e\text{-}4}$ | $6.4_{\pm 8e\text{-}4}$ | $6.3_{\pm 7e\text{-}4}$ |
| Threshold | $31.8_{\pm 2e\text{-}3}$ | $28.8_{\pm 2e\text{-}3}$ | $27.1_{\pm 1e\text{-}3}$ | $26.8_{\pm 2e\text{-}3}$ | $26.8_{\pm 7e\text{-}4}$ | $26.8_{\pm 7e\text{-}4}$ | $26.8_{\pm 7e\text{-}4}$ |

and directly use the ensemble covariance $\boldsymbol{A}_O \equiv \boldsymbol{\Sigma}_x$ as observation. As before, we take a training/validation split of 913/500. Figure 3 shows GDNs effectively generalize to graphs orders of magnitude larger than they were trained on, giving up only modest levels of performance as size increases. Note that the best performing baseline in Table 1 - trained *and tested* on the original $N = 68$ domain - is not comparable with GDN-S in terms of performance until GDN-S is tested on graphs almost an order of magnitude larger than those it was trained on. The GDN-S model showed a more graceful performance degradation, suggesting that GDNs without parameter sharing may be using their extra representational power to pick up on finite-size effects, which may disappear as $N$ increases. The shared parameter constraint acts as regularization, we avoid over-fitting on a given size domain to better generalize to larger graphs.

**Ablation studies.** The choice of prior can influence model performance, as well as reduce training time and the number of parameters needed. When run on stochastic block model (SBM) graphs with $N = 21$ nodes and 3 equally-sized communities (within block connection probability of 0.6, and 0.1 across blocks), for the link prediction task GDNs attain an error of $16.8 \pm 2.7e\text{-}2\%$, $16.0 \pm 2.1e\text{-}2\%$, $14.5 \pm 1.0e\text{-}2\%$, $14.3 \pm 8.8e\text{-}2\%$ using a zeros, ones, block diagonal (using the SBM edge connectivity probabilities), and learned prior, respectively. The performance improves when GDNs are given an informative prior (here a block diagonal matrix matching the graph communities), with further gains when GDNs are allowed to learn $\boldsymbol{A}[0]$.

We also study the effect of gradient truncation. To derive GDNs we approximate the gradient $\nabla g(\boldsymbol{A})$ by dropping all higher-order terms in $\boldsymbol{A}$ ($K = 1$). The case of $K = 0$ corresponds to further dropping the terms linear in $\boldsymbol{A}$, leading to PG iterations $\boldsymbol{A}[k+1] = \mathrm{ReLU}(\boldsymbol{A}[k] + \gamma \boldsymbol{A}_O - \tau \mathbf{1}\mathbf{1}^\top)$ [cf. eq. (6)]. We run this simplified iterative soft-thresholding model with $D = 8$ layers and $C = 8$ channels per layer on the same RG graphs in Table 1. Lacking the linear term that facilitates information aggregation in the graph, the model is not expressive enough and yields a higher error (MSE) of $25.7 \pm 1.3e\text{-}2\%$ ($1.7e\text{-}1 \pm 4.7e\text{-}4$) for the link-prediction (edge weight regression) task. Models with $K \geq 2$ result in unstable training with highly fluctuating loss values, which we attribute to a challenging optimization landscape as polynomial minimization is known to be notoriously hard, leading to less well-behaved gradients. This motivates our choice of $K = 1$ in GDNs, which are nonetheless flexible enough to capture a rich class of (higher-order polynomial) convolutional processes by stacking multiple linear layers as in Figure 1.

## 5.2 Real Data

**HCP-YA neuroimaging dataset.** HCP represents a unifying paradigm to acquire high quality neuroimaging data across studies that enabled unprecedented quality of macro-scale human brain connectomes for analyses in different domains (Glasser et al., 2016). We use the dMRI and resting state fMRI data from the HCP-YA dataset (Van Essen et al., 2013), consisting of 1200 healthy, young adults (ages: 22-36 years). The SC and FC are projected on a common brain atlas, which is a grouping of cortical structures in the brain to distinct regions. We interpret these regions as nodes in a brain graph. For our experiments, we use the standard Desikan-Killiany atlas (Desikan et al., 2006) with $N = 68$ cortical brain regions. The SC-FC coupling on this dataset is known to be the strongest in the occipital lobe and vary with age, sex and cognitive health in other subnetworks (Gu et al., 2021). Under the consideration of the variability in SC-FC coupling across the brain regions, we further group the cortical regions into 4 neurologically defined 'lobes': frontal, parietal, temporal, and occipital (the locations of these lobes in the brain are included in Fig. 6 in Appendix A.6). We predict SC, derived from dMRI, using FC, constructed using BOLD signals acquired with fMRI. In the literature, the forward (SC to FC) problem has been mostly attempted, a non-trivial yet simpler task due to the known reliance of function on structure in many cortical regions. We tackle the markedly harder

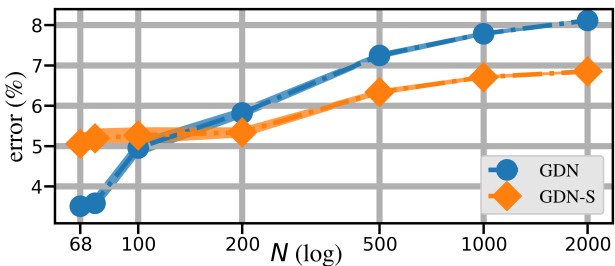

Figure 3: Size Generalization: GDNs maintain performance on RG graphs orders of magnitude larger than the $N = 68$ training set.

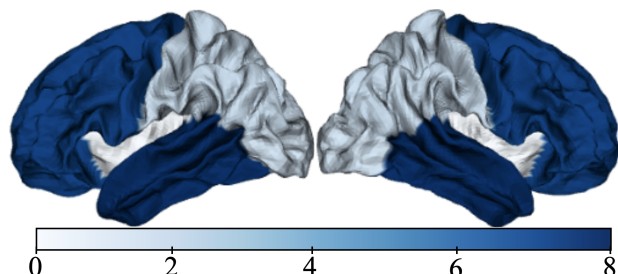

Figure 4: Reduction in MAE (%) for different lobes; largest improvements occur in temporal and frontal lobes.

inverse (deconvolution) problem of recovering SC from FC, which has so far received less attention and can have major impact as described in Section 3. Notably, this network deconvolution problem represents an ideal fit to the polynomial model (1) relating FC with SC (Liang & Wang, 2017; Abdelnour et al., 2014), plus the supervised setting is well motivated given the availability of functional and structural connectomes from the HCP-YA repository.

From this data, we extracted a dataset of 1063 FC-SC pairs, $\mathcal{T} = \{\boldsymbol{FC}^{(i)}, \boldsymbol{SC}^{(i)}\}_{i=1}^{1063}$ and use a training/validation/test split of 913/50/100. Taking the prior $\boldsymbol{A}[0]$ as the edgewise mean over all SCs in the training split $\mathcal{T}_{train}$: $A[0]_{ij} = \text{mean} \{SC_{i,j}^{(1)}, \ldots, SC_{i,j}^{(913)}\}$ and using it directly as a predictor on the test set (tuning a threshold with validation data), we achieve strong performance on link-prediction and edge-weight regression tasks on the whole brain (error = 12.3%, MAE = 0.0615). At the lobe level, the prior achieves relatively higher accuracy in occipital (error = 1.3%, MAE = 0.064) and parietal (error = 6.6%, MAE = 0.07) lobes as compared to temporal (error = 11.0%, MAE = 0.053) and frontal (error = 10.5%, MAE = 0.062) lobes; behavior which is unsurprising as SC in temporal and frontal lobes are affected by ageing and gender related variability in the dataset (Zimmermann et al., 2016). GDNs reduced MAE by 7.6%, 7.1%, 1.6%, and 1.3% in the temporal, frontal, parietal, and occipital networks respectively and 8.0% over the entire brain network, all relative to the already strong mean prior. The four lobe reductions are visualized in Figure 4. We observed the most significant gains over temporal and frontal lobes; clearly there was smaller room to improve performance over the occipital and frontal lobes. The resulting MAEs for the entire brain network for L2G, GLAD, GCN-D, and GIN-D are 2.23e-1, 2.07e-1, 2.23e-1, 2.10e-1 respectively; none were able to reduce MAE (or error).

In summary, our 'pseudo-synthetic' experiments in Section 5.1 show that SCs are amenable to learning with GDNs when the SC-FC relationship satisfies eq. (1), a reasonable model given the findings of (Abdelnour et al., 2014; Liang & Wang, 2017). In general, SC-FC coupling can vary widely across both the population and within the sub-regions of an individuals brain for healthy subjects and in pathological contexts. When trained on the HCP-YA dataset, GDNs exhibit robust performance over such regions with high variability in SC-FC coupling. Therefore, our results on HCP-YA dataset justify GDNs as the leading baseline model for comparisons in this significant FC to SC task. It could also potentially serve as a baseline for characterizing healthy subjects in pathology studies in future work, where pathology could be characterized by specific deviations from our results.

**Friendship recommendation from physical co-location networks.** Here we use GDNs to predict Facebook ties given human co-location data. GDNs are well suited for this deconvolution problem since one can view friendships as direct ties, whereas co-location edges include indirect relationships due to casual encounters in addition to face-to-face contacts with friends. In terms of impact, a trained model could then be useful to augment friendship recommendation engines given co-location (behavioral) data. Granted, this would also pose interesting privacy questions that are beyond the scope of this paper, and certainly call for tighter control of co-location information.

The Thiers13 dataset (Génois & Barrat, 2018) monitored high school students, recording: (i) physical co-location interactions with wearable sensors over 5 days; and (ii) social network information via survey.

From this we construct a dataset $\mathcal{T}$ of graph pairs, each with $N = 120$ nodes, where $\boldsymbol{A}_O$ are co-location networks, i.e., weighted graphs where the weight of edge $(i, j)$ represents the number of times student $i$ and $j$ came into physical proximity, and $\boldsymbol{A}_L$ is the Facebook subgraph between the same students; additional data pre-processing details are in Appendix A.6. We trained a GDN of depth $D = 11$ without MIMO filters ($C = 1$), with learned $\boldsymbol{A}[0]$ using a training/validation/test split of 5000/1000/1000. We achieved a test error of $8.9 \pm 1.5e\text{-}2\%$, a 13.0% reduction over next best performing baseline, suggesting we learn a latent mechanism that the baselines (even GLAD, a supervised graph learning method) cannot. The performance of the baselines are as follows: Threshold ($10.2 \pm 1.2e\text{-}2\%$), ND ($10.2 \pm 1.2e\text{-}2\%$), GLASSO (NA - could not converge for any of the $\alpha$ values tested), SpecTemp ($10.7 \pm 6.9e\text{-}2\%$), GLAD ($14.7 \pm 1.3e\text{-}2\%$), GCN-C ($11.7 \pm 3.7e\text{-}4\%$), and GIN-D ($11.5 \pm 4.0e\text{-}4\%$). See Appendix A.6 for further details.

## 6 Conclusions

In this work we proposed the GDN, an inductive model capable of recovering latent graph structure from observations of its convolutional mixtures. Bringing to bear the principle of algorithm unrolling, we introduce a novel deep learning solution to a network inverse (deconvolution) problem in a supervised fashion, that is well motivated e.g., to predict whole brain structural connectivity from functional signals. By minimizing a task-dependent loss function, GDNs learn filters to refine initial estimates of the sought latent graphs layer by layer. The unrolled NN can seamlessly integrate domain-specific prior information about the unknown graph distribution. GDNs enjoy a node permutation equivariance property, which is key to ensure the learnt graph topologies are not affected by arbitrary vertex labelings. Moreover, because GDNs: (i) are differentiable functions with respect to their parameters as well as their graph input; and (ii) offer explicit control on complexity (leading to fast inference times); one can envision GDNs as valuable components in larger (even online) end-to-end graph representation learning systems. This way, while our focus here has been exclusively on for graph structure learning, the impact of GDNs can permeate to broader graph inference tasks.

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

## A Appendix

### A.1 Model Identifiability

Without any constraints on $\boldsymbol{h}$ and $\boldsymbol{A}_L$, the problem of recovering $\boldsymbol{A}_L$ from $\boldsymbol{A}_O = \boldsymbol{H}(\boldsymbol{A}_L; \boldsymbol{h})$ as in eq.(1) is clearly non-identifiable. Indeed, if the desired solution is $\boldsymbol{A}_L$ (with associated polynomial coefficients $\boldsymbol{h}$), there is always at least another solution $\mathbf{A}_O$ corresponding to the identity polynomial mapping. This is why adding structural constraints like sparsity on $\boldsymbol{A}_L$ will aid model identifiability, especially when devoid of training examples.

It is worth mentioning that eq. (1) implies the eigenvectors of $\boldsymbol{A}_L$ and $\boldsymbol{A}_O$ coincide. So the eigenvectors of the sought latent graph are given once we observe $\boldsymbol{A}_O$, what is left to determine are the eigenvalues. We have in practice observed that for several families of sparse, weighted graphs, the eigenvector information along with the constraint $\boldsymbol{A}_L \in \mathcal{A}$ are sufficient to uniquely specify the graph. Interestingly, this implies that many random weighted graphs can be uniquely determined from their eigenvectors. This strong uniqueness result does not render our problem vacuous, since seldomly in practice one gets to observe $\boldsymbol{A}_O$ (and hence its eigenvectors) error free.

If one were to formally study identifiability of eq. (1) (say under some structural assumptions on $\boldsymbol{A}_L$ and/or the polynomial mapping), then one has to recognize the problem suffers from an inherent scaling ambiguity.

Indeed, if given $\boldsymbol{A}_O = \boldsymbol{H}(\boldsymbol{A}_L; \boldsymbol{h})$ which means the pair $\boldsymbol{A}_L$ and $\boldsymbol{h} = [h_0, h_1, \ldots, h_K]^\top$ is a solution, then for any positive scalar $\alpha$ one has that $\alpha \boldsymbol{A}_L$ and $[h_0, h_1/\alpha, \ldots, h_K/(\alpha^K)]^\top$ is another solution. Accordingly, uniqueness claims can only be meaningful modulo this unavoidable scaling ambiguity. But this ambiguity is lifted once we tackle the problem in a supervised learning fashion – our approach in this paper. The training samples in $\mathcal{T} := \{\boldsymbol{A}_O^{(i)}, \boldsymbol{A}_L^{(i)}\}_{i=1}^T$ fix the scaling, and accordingly the GDN can learn the mechanism or mapping of interest $\boldsymbol{A}_O \mapsto \boldsymbol{A}_L$. Hence, an attractive feature of the GDN approach is that by using data, some of the inherent ambiguities in eq. (1) are naturally overcome. In particular, the SpecTemp approach in (Segarra et al., 2017) relies on convex optimization and suffers from this scaling ambiguity, so it requires an extra (rather arbitrary) constraint to fix the scale. The network deconvolution approach (ND) in (Feizi et al., 2013) relies on a fixed, known polynomial mapping, and while it does not suffer from these ambiguities it is limited in the graph convolutions it can model.

All in all, the inverse problem associated to eq. (1) is just our starting point to motivate a trainable parametrized architecture $\hat{\boldsymbol{A}}_L = \Phi(\boldsymbol{A}_O; \boldsymbol{\Theta})$, that introduces valuable inductive biases to generate graph predictions. The problem we end up solving is different (recall the formal statement in Section 2) because we rely on supervision using graph examples, thus rendering many of these challenging uniqueness questions less relevant.

## A.2 Technical Details of the Proximal Gradient Algorithm

**Derivation of the proximal gradient iterations (4).** Here we derive the proximal gradient (PG) algorithm for the composite optimization problem (3). With $\alpha > 0$ and $\mathcal{A}$ a convex set, introduce the proximal operator of a function $\alpha\varphi(\cdot) : \mathbb{R}^{N \times N} \to \mathbb{R}$ evaluated at matrix $\boldsymbol{M} \in \mathbb{R}^{N \times N}$ as

$$\boldsymbol{Z}(\boldsymbol{M}) = \text{prox}_{\alpha\varphi, \mathcal{A}}(\boldsymbol{M}) := \underset{\boldsymbol{X} \in \mathcal{A}}{\arg\min} \left[ \varphi(\boldsymbol{X}) + \frac{1}{2\alpha} \|\boldsymbol{X} - \boldsymbol{M}\|_F^2 \right]. \tag{8}$$

With these definitions, the PG updates with fixed step-size $\tau$ to solve the network inverse problem (3) are given by ($k = 0, 1, 2, \ldots$ denote iterations)

$$\boldsymbol{A}[k+1] := \text{prox}_{\tau\|\cdot\|_1, \mathcal{A}} \left( \boldsymbol{A}[k] - \tau \nabla g(\boldsymbol{A}[k]) \right). \tag{9}$$

Notice that $\tau$ should be appropriately chosen as a function of the regularization parameter $\lambda$ in (3).

Evaluating the proximal operator efficiently is key to the success of PG methods. For our specific case of sparse graph learning where $\varphi(\boldsymbol{A}) = \|\boldsymbol{A}\|_1$ and the optimization variable is constrained as $\boldsymbol{A} \in \mathcal{A}$, the proximal operator $\boldsymbol{Z}$ in (8) has entries given by

$$Z_{ij}(M_{ij}) = \begin{cases} 0, & i = j \\ \max(0, M_{ij} - \alpha), & \text{otherwise.} \end{cases} \tag{10}$$

The resulting entry-wise separable nonlinear map nulls the diagonal entries of $\boldsymbol{A}[k+1]$, and applies a non-negative soft-thresholding operator to update the remaining entries. Recognizing the $\alpha$-shifted rectified linear unit $\text{ReLU}(M_{ij} - \alpha) = \max(0, M_{ij} - \alpha)$ and substituting the proximal operator expression (10) in (9), one arrives at the PG iterations (4). We recall that as per (10), the matrix-valued ReLU operator in (4 sets $\text{diag}(\boldsymbol{A}[k+1]) = \boldsymbol{0}$. If $\boldsymbol{A}[0] \in \mathcal{A}$, then symmetry of all subsequent iterates $\boldsymbol{A}[k+1]$, $k = 0, 1, 2, \ldots$ will be preserved.

**Gradient calculation.** Here we give mathematical details in the calculation of the gradient $\nabla g(\boldsymbol{A})$ of the component function $g(\boldsymbol{A}) := \frac{1}{2}\|\boldsymbol{A}_O - \boldsymbol{H}(\boldsymbol{A}; \boldsymbol{h})\|_F^2$. Let $\boldsymbol{A}, \boldsymbol{A}_O$ be symmetric $N \times N$ matrices and recall the

graph filter $\boldsymbol{H}(\boldsymbol{A}) := \sum_{k=0}^{K} h_k \boldsymbol{A}^k$ (we drop the dependency in $\boldsymbol{h}$ to simply the notation). Then

$$
\begin{aligned}
\nabla_{\boldsymbol{A}} \frac{1}{2} \|\boldsymbol{A}_O - \boldsymbol{H}(\boldsymbol{A})\|_F^2 &= \frac{1}{2} \nabla_{\boldsymbol{A}} \operatorname{Tr} \left( \boldsymbol{A}_O^2 - \boldsymbol{A}_O \boldsymbol{H}(\boldsymbol{A}) - \boldsymbol{H}(\boldsymbol{A}) \boldsymbol{A}_O + \boldsymbol{H}^2(\boldsymbol{A}) \right) \\
&= -\nabla_{\boldsymbol{A}} \operatorname{Tr} \left( \boldsymbol{A}_O \boldsymbol{H}(\boldsymbol{A}) \right) + \frac{1}{2} \nabla_{\boldsymbol{A}} \operatorname{Tr} \boldsymbol{H}^2(\boldsymbol{A}) \\
&= -\sum_{k=1}^{K} h_k \sum_{r=0}^{k-1} \boldsymbol{A}^{k-r-1} \boldsymbol{A}_O \boldsymbol{A}^r + \frac{1}{2} \nabla_{\boldsymbol{A}} \operatorname{Tr} \boldsymbol{H}^2(\boldsymbol{A}) \\
&= -\sum_{k=1}^{K} h_k \sum_{r=0}^{k-1} \boldsymbol{A}^{k-r-1} \boldsymbol{A}_O \boldsymbol{A}^r + \frac{1}{2} \boldsymbol{H}_1(\boldsymbol{A}) \\
&= -[h_1 \boldsymbol{A}_O + h_2 (\boldsymbol{A} \boldsymbol{A}_O + \boldsymbol{A}_O \boldsymbol{A}) + h_3 (\boldsymbol{A}^2 \boldsymbol{A}_O + \boldsymbol{A} \boldsymbol{A}_O \boldsymbol{A} + \boldsymbol{A}_O \boldsymbol{A}^2) + \ldots] + \frac{1}{2} \boldsymbol{H}_1(\boldsymbol{A}),
\end{aligned}
$$

where in arriving at the second equality we relied on the cyclic property of the trace, and $\boldsymbol{H}_1(\boldsymbol{A})$ is a matrix polynomial of order $2K - 1$.

Note that in the context of the GDN model, powers of $\boldsymbol{A}$ will lead to complex optimization landscapes, and thus unstable training. We thus opt to drop the higher-order terms and work with a first-order approximation of $\nabla g$, namely

$$
\nabla g(\boldsymbol{A}) \approx -h_1 \boldsymbol{A}_O - h_2 (\boldsymbol{A}_O \boldsymbol{A} + \boldsymbol{A} \boldsymbol{A}_O) + (2 h_0 h_2 + h_1^2) \boldsymbol{A}.
$$

### A.3 Incorporating Prior Information

In Section 4.3 we introduce the concept of using prior information in the training of GDNs. We do so by encoding information we may have about the unknown latent graph $\boldsymbol{A}_L$ into $\boldsymbol{A}[0]$, the initial state which GDNs iteratively refine. If the $\boldsymbol{A}_L$'s are repeated instances of a graph with fixed nodes, as is the case with the SCs with respect to the 68 fixed brain regions, a simple strategy to incorporate prior information about some edge $\mathbf{A}_{L_{i,j}}$, now viewed as a random variable, would be $\boldsymbol{A}[0]_{i,j} \leftarrow \mathbb{E}[\mathbf{A}_{L_{i,j}}]$. But there is more that can be done. We also can estimate the variance $\operatorname{Var}(\mathbf{A}_{L_{i,j}})$, and use it during the training of a GDN, for example taking $\boldsymbol{A}[0]_{i,j} \leftarrow \mathcal{N}(\mathbb{E}[\mathbf{A}_{L_{i,j}}], \operatorname{Var}(\mathbf{A}_{L_{i,j}}))$, or even simpler, using a resampling technique and taking $\boldsymbol{A}[0]_{i,j}$ to be a random sample in the training set. By doing so, we force the GDN to take into account the distribution and uncertainty in the data, possibly leading to richer learned representations and better performance. It also would act as a form of regularization, not allowing the model to converge on the naive solution of outputting the simple expectation prior, a likely local minimum in training space. We also can learn the prior $\boldsymbol{A}[0]$ by treating all entries as parameters directly; note due to undirectedness and lack of self-loops this would require $N(N-1)/2$ additional parameters.

### A.4 MIMO Model Architecture

**MIMO filters.** Formally, the more expressive GDN architecture with MIMO (Multi-Input Multi-Output) filters is constructed as follows. At layer $k$ of the NN, we take a three-way tensor $\mathbf{A}_k \in \mathbb{R}_+^{C \times N \times N}$ and produce $\mathbf{A}_{k+1} \in \mathbb{R}_+^{C \times N \times N}$, where $C$ is the common number of input and output channels. The assumption of having a common number of input and output channels can be relaxed, as we argue below. By defining multiplication between tensors $\mathbf{T}, \mathbf{B} \in \mathbb{R}^{C \times N \times N}$ as batched matrix multiplication: $[\mathbf{T}\mathbf{B}]_{j,:,:} := \mathbf{T}_{j,:,:} \mathbf{B}_{j,:,:}$, and tensor-vector addition and multiplication as $\mathbf{T} + \boldsymbol{v} := \mathbf{T} + [v_1 \mathbf{1} \mathbf{1}^\top; \ldots; v_C \mathbf{1} \mathbf{1}^\top]$ and $[\boldsymbol{v} \mathbf{T}]_{j,:,:} := v_j \mathbf{T}_{j,:,:}$ respectively for $\boldsymbol{v} \in \mathbb{R}^C$, all operations extend naturally.

Using these definitions, the $j$-th output slice of layer $k$ is

$$
[\mathbf{A}_{k+1}]_{j,:,:} = \operatorname{ReLU}[\overline{\boldsymbol{\alpha}_{:,j} \mathbf{A}_k + \boldsymbol{\beta}_{:,j} (\mathbf{A}_O \mathbf{A}_k + \mathbf{A}_k \mathbf{A}_O) + \boldsymbol{\gamma}_{:,j} \mathbf{A}_O} - \tau_j \mathbf{1} \mathbf{1}^\top], \tag{11}
$$

where $\overline{\cdots}$ represents the mean reduction over the filtered input channels and the parameters are $\boldsymbol{\alpha}, \boldsymbol{\beta}, \boldsymbol{\gamma} \in \mathbb{R}^{C \times C}$, $\boldsymbol{\tau} \in \mathbb{R}_+^C$. We now take $\boldsymbol{\Theta} := \{\boldsymbol{\alpha}, \boldsymbol{\beta}, \boldsymbol{\gamma}, \boldsymbol{\tau}\}$ for a total of $C \times (3C + 1)$ trainable parameters.

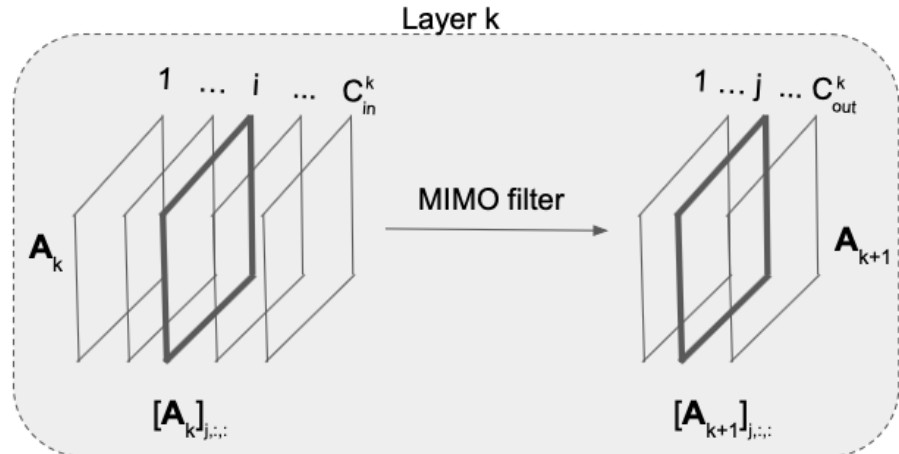

Figure 5: MIMO Filter: Layer $k$ takes a tensor $\mathbf{A}_k \in \mathbb{R}^{C_{in}^k \times N \times N}$ and outputs a tensor $\mathbf{A}_{k+1} \in \mathbb{R}^{C_{out}^k \times N \times N}$. The $i$-th slice $[\mathbf{A}_k]_{i,:,:}$ is called the $i$-th input channel and $[\mathbf{A}_{k+1}]_{j,:,:}$ is called the $j$-th output channel.

We typically have a single prior matrix $\boldsymbol{A}[0]$ and are interested in predicting a single adjacency matrix $\hat{\boldsymbol{A}}_L$. Accordingly, we construct a new tensor prior $\mathbf{A}[0] := [\boldsymbol{A}[0], \ldots, \boldsymbol{A}[0]] \in \mathbb{R}^{C \times N \times N}$ and (arbitrarily) designate the first output channel as our prediction $\hat{\boldsymbol{A}}_L = \Phi(\boldsymbol{A}_O; \boldsymbol{\Theta}) = [\mathbf{A}_{k+1}]_{1,:,:}$.

We can also allow each layer to learn a decoupled MIMO filter, with its own set of parameters mapping from $C_{in}^k$ input channels to $C_{out}^k$ output channels. As the notation suggests, $C_{in}^k$ and $C_{out}^k$ need not be equal. Layer $k$ now has its own set of parameters $\boldsymbol{\Theta}^k = (\boldsymbol{\alpha}^k, \boldsymbol{\beta}^k, \boldsymbol{\gamma}^k, \boldsymbol{\tau}^k)$, where $\boldsymbol{\alpha}^k, \boldsymbol{\beta}^k, \boldsymbol{\gamma}^k \in \mathbb{R}^{C_{out}^k \times C_{in}^k}$ and $\boldsymbol{\tau}^k \in \mathbb{R}_+^{C_{out}^k}$, for a total of $C_{out}^k \times (3C_{in}^k + 1)$ trainable parameters. The tensor operations mapping inputs to outputs remains basically unchanged with respect to (11), except that the filter coefficients will depend on $k$.

Processing at a generic layer $k$ is depicted in Figure 5. Output channel $j$ will use $\boldsymbol{\alpha}_{:,j}^k, \boldsymbol{\beta}_{:,j}^k, \boldsymbol{\gamma}_{:,j}^k \in \mathbb{R}^{C_{in}^k}$ and $\tau_j^k \in \mathbb{R}_+$ to filter all input channels $i \in \{1, \cdots, C_{in}^k\}$, which are collected in the input tensor $\mathbf{A}_k \in \mathbb{R}^{C_{in}^k \times N \times N}$. This produces a tensor of stacked filtered input channels $\in \mathbb{R}^{C_{in}^k \times N \times N}$. After setting the diagonal elements of all matrix slices in this tensor to 0, then perform a mean reduction edgewise (over the first mode/dimension) of this tensor, producing a single $N \times N$ matrix. We then apply two pointwise/elementwise operations on this matrix: (i) subtract $\boldsymbol{\tau}_j^k$ (this would be the 'bias' term in CNNs); and (ii) apply a point-wise nonlinearity (ReLU). This produces an $N \times N$ activation stored in the $j$-th output channel. Doing so for all output channels $j \in \{1, \cdots, C_{out}^k\}$, produces a tensor $\mathbf{A}_{k+1} \in \mathbb{R}^{C_{out}^k \times N \times N}$.

**Layer output normalization and zeroing diagonal.** The steps not shown in the main text are the normalization steps, a practical issue, and the setting of the diagonal elements to be 0, a projection onto the set $\mathcal{A}$ of allowable adjacency matrices. Define $\overline{\mathbf{U}}_{j,:,:} = \overline{\boldsymbol{\alpha}_{:,j}^k \mathbf{A}_k + \boldsymbol{\beta}_{:,j}^k (\mathbf{A}_O \mathbf{A}_k + \mathbf{A}_k \mathbf{A}_O) + \boldsymbol{\gamma}_{:,j}^k \mathbf{A}_O} \in \mathbb{R}^{N \times N}$ as the $j$-th slice in the intermediate tensor $\overline{\mathbf{U}} \in \mathbb{R}^{C_{out}^k \times N \times N}$ used in the filtering of $\mathbf{A}_k$. Normalization is performed by dividing each matrix slice of $\overline{\mathbf{U}}$ by the maximum magnitude element in that respective slice: $\overline{\mathbf{U}}_{\cdot,:,:} / \max \left| \overline{\mathbf{U}}_{\cdot,:,:} \right|$.

Multiple normalization metrics were tried on the denominator, including the $99th$ percentile of all values in $\overline{\mathbf{U}}_{\cdot,:,:}$, the Frobenius norm $\|\overline{\mathbf{U}}_{\cdot,:,:}\|_F$, among others. None seemed to work as well as the maximum magnitude element, which has the additional advantage of guaranteeing entries to be in $[0, 1]$ (after ReLU), which matches nicely with: (i) adjacency matrices of unweighted graphs; and (ii) makes it easy to normalize edge weights of a dataset of adjacencies: simply scale them to $[0, 1]$.

In summary, the full procedure to produce $[\mathbf{A}_{k+1}]_{j,:,:}$ is as follows:

$$\overline{\mathbf{U}}_{j,:,:} = \overline{\boldsymbol{\alpha}_{:,j}^k \mathbf{A}_k + \boldsymbol{\beta}_{:,j}^k (\mathbf{A}_O \mathbf{A}_k + \mathbf{A}_k \mathbf{A}_O) + \boldsymbol{\gamma}_{j,:}^k) \mathbf{A}_O}$$

$$\overline{\mathbf{U}}_{j,:,:} = \overline{\mathbf{U}}_{j,:,:} \odot (\mathbf{1}\mathbf{1}^\top - \boldsymbol{I}) \qquad \text{force diagonal elements to 0}$$

$$\overline{\mathbf{U}}_{j,:,:} = \overline{\mathbf{U}}_{j,:,:} / \max(|\overline{\mathbf{U}}_{j,:,:}|) \qquad \text{normalize entries per slice to be in } [-1, 1]$$

$$[\mathbf{A}_{k+1}]_{j,:,:} = \mathrm{ReLU}(\overline{\mathbf{U}}_{j,:,:} - \tau_l^j)$$

By normalizing in this way, we guarantee the intermediate matrix $\overline{\mathbf{U}}_{j,:,:}$ has entries in $[-1, 1]$ (before the ReLU). This plays two important roles. The first one has to do with training stability and to appreciate this point consider what could happen if no normalization is used. Suppose the entries of $\overline{\mathbf{U}}_{j,:,:}$ are orders of magnitude larger than entries of $\overline{\mathbf{U}}_{l,:,:}$. This can cause the model to push $\tau_j^k >> \tau_l^k$, leading to training instabilities and/or lack of convergence. The second point relates to interpretability of $\tau$. Indeed, the proposed normalization allows us to interpret the learned values $\boldsymbol{\tau}^k \in \mathbb{R}_{out,+}^k$ on a similar scale. All the thresholds must be in $[0, 1]$ because: (i) anything above 1 will make the output all 0; and (ii) we constrain it to be non-negative. In fact we can now plot all $\tau$ values (from all layers) against one another, and using the same scale ($[0, 1]$) interpret if a particular $\tau$ is promoting a lot of sparsity in the output ($\tau$ close to 1) or not ($\tau$ close to 0), by examining its magnitude.

**Gaining expressiveness by generalizing the channel reduction.** MIMO filters provide a significant boost in performance, and effectively scale well to large networks due to their reliance on a minimal set of fast operations such as batched (sparse) matrix multiplication and addition, as well as their extreme parameter efficiency. This is in contrast to recent highly parametrized models e.g. Graph Transformers (Dwivedi et al., 2022), which require significantly more computational resources - effectively limiting their application to very small networks. Still, more expressiveness may be desired, and one such solution discussed below fits elegantly into the MIMO formulation of GDNs and still has nice scaling properties.

Consider a layer $L$ which takes an input tensor $\overline{\mathbf{T}}_{in} \in \mathbb{R}^{C_{in} \times N \times N}$ and outputs tensor $\overline{\mathbf{T}}_{out} \in \mathbb{R}^{C_{out} \times N \times N}$. A single channel $i$ in the output filters $\overline{\mathbf{T}}_{in}$ to construct it's own filtered tensor of identical size. It then performs a mean/sum reduction over the first dimension of this constructed tensor, producing a single slice in the output tensor $\overline{\mathbf{T}}_{out}[i] \in \mathbb{R}^{N \times N}$. Note that this reduction essentially acts on the filtered 'edge features' that the output channel $i$ produced. To gain expressiveness, this reduction transformation on the edge features can be swapped out for a more expressive transformation, e.g. an MLP, which is shared across all edges in the single output channel, and optionally all output channels within the layer (and optionally all layers in the network) - reminiscent of the parameter sharing used in GNNs to transform node features. This sharing is what provides advantageous scaling properties.

**Incorporating node features.** In this work, we focus on the network inverse problem introduced in Section 2. In this problem setting, we observe a noisy graph $\boldsymbol{A}_O \in \mathbb{R}^{N \times N}$ and look to recover the true latent graph $\boldsymbol{A}_L \in \mathbb{R}^{N \times N}$; the problem does not consider the use of node features, and thus GDNs do not directly consider them. But if they are available, GDNs can readily leverage them with the use of MIMO filters and link-prediction techniques which are standard in the DL literature; see the ensuing discussion for details. Given two nodes $i, j \in \mathcal{V}$ with respective node features $\boldsymbol{v}_i, \boldsymbol{v}_j \in \mathbb{R}^m$, define a nonlinear function $h_\Theta : \mathbb{R}^m \times \mathbb{R}^m \to \mathbb{R}^{m'}$ with set of learnable parameters $\Theta$. As is standard in the Geometric Deep Learning literature (Bronstein et al., 2017), we can feed $h_\Theta$ any elementwise symmetric function of $\boldsymbol{v}_i, \boldsymbol{v}_j$ - e.g. the elementwise squared difference $(\boldsymbol{v}_i - \boldsymbol{v}_j) \odot (\boldsymbol{v}_i - \boldsymbol{v}_j)$, elementwise Gaussian kernel $g$ of the difference $g(\boldsymbol{v}_i - \boldsymbol{v}_j)$, etc. We organize the pairwise outputs of $h_\Theta$ into a tensor $\overline{\mathbf{R}} \in \mathbb{R}^{m' \times N \times N}$ such that $\overline{\mathbf{R}}_{:,i,j} = \overline{\mathbf{R}}_{:,j,i} = h_\Theta(\boldsymbol{v}_i, \boldsymbol{v}_j)$. In the MIMO formulation, the first GDN layer already has a set of input channels $\overline{\mathbf{T}}_{in}$ (some slices of $\overline{\mathbf{T}}_{in}$ can be prior information, learned, or zeros); we can simply append this new tensor $\overline{\mathbf{R}}$ as more slices in $\overline{\mathbf{T}}_{in}$ and proceed normally.

We leave these extensions for future work.

## A.5 Training

Training of the GDN model will be performed using stochastic (mini-batch) gradient descent to minimize a task-dependent loss function $L(\boldsymbol{\Theta})$ as in (2). The loss is defined either as (i) the edgewise squared/absolute error between the predicted graph and the true graph for regression tasks, or (ii) a hinge loss with parameter $\gamma \geq 0$, both averaged over a training set $\mathcal{T} := \{\boldsymbol{A}_O^{(i)}, \boldsymbol{A}_L^{(i)}\}_{i=1}^T$, namely

$$
\ell_{\text{hinge}}(\boldsymbol{A}_L^{(i)}, \Phi(\boldsymbol{A}_O^{(i)}; \boldsymbol{\Theta})) := \sum_{i,j} \begin{cases} (\Phi(\boldsymbol{A}_O^{(i)}; \boldsymbol{\Theta})_{i,j} - \gamma)^+ & \boldsymbol{A}_{L_{i,j}} = 0 \\ (-\Phi(\boldsymbol{A}_O^{(i)}; \boldsymbol{\Theta})_{i,j} + 1 - \gamma)^+ & \boldsymbol{A}_{L_{i,j}} > 0 \end{cases},
$$

$$
\ell_{\text{mse}}(\boldsymbol{A}_L^{(i)}, \Phi(\boldsymbol{A}_O^{(i)}; \boldsymbol{\Theta})) := \frac{1}{2} \left\| \boldsymbol{A}_L^{(i)} - \Phi(\boldsymbol{A}_O^{(i)}; \boldsymbol{\Theta}) \right\|_2^2,
$$

$$
\ell_{\text{mae}}(\boldsymbol{A}_L^{(i)}, \Phi(\boldsymbol{A}_O^{(i)}; \boldsymbol{\Theta})) := \left\| \boldsymbol{A}_L^{(i)} - \Phi(\boldsymbol{A}_O^{(i)}; \boldsymbol{\Theta}) \right\|_1,
$$

$$
L(\boldsymbol{\Theta}) := \frac{1}{T} \sum_{i \in \mathcal{T}} \ell_u(\boldsymbol{A}_L^{(i)}, \Phi(\boldsymbol{A}_O^{(i)}; \boldsymbol{\Theta})), \qquad u \in \{\text{hinge, mse, mae}\}.
$$

The loss is optimized with ADAM using a learning rate of 0.01, $\beta_1 = 0.85$, and $\beta_2 = 0.99$.

**Link prediction with GDNs and unbiased estimates of generalization.** In the edge-weight regression task, GDNs only use their validation data to determine when training has converged. When performing link-prediction, GDNs have an additional use for this data: to choose the cutoff threshold $t \in \mathbb{R}_+$, determining which raw outputs (which are continuous) should be considered positive edge predictions, *at the end of training.*

We use the training set to learn the parameters (via gradient descent) *and* to tune $t$. During the validation step, when then use this train-set-tuned-$t$ on the validation data, giving an estimate of generalization error. This is then used for early-stopping, determining the best model learned after training, etc. We do not use the validation data to tune $t$ during training. Only after training has completed, do we tune $t$ with validation data. We train a handful of models this way, and the model which produces the best validation score (in this case lowest error) is the tested with the validation-tuned-$t$, thus providing an unbiased estimate of generalization.

## A.6 Notes on the Experimental Setup

**Synthetic graphs.** For the experiments presented in Table 1, the synthetic graphs of size $N = 68$ are drawn from random graph models with the following parameters

- Random geometric graphs (RG): $d = 2$, $r = 0.56$.
- Erdős-Rényi (ER): $p = .56$.
- Barabási-Albert (BA): $m = 15$

When sampling graphs to construct the datasets, we reject any samples which are not connected or have edge density outside of a given range. For RG and ER, that range is $[0.5, 0.6]$, while in BA the range is $[0.3, 0.4]$. This is an attempt to make the RG and ER graphs similar to the brain SC graphs, which have an average edge density of 0.56, and all SCs are in edge density range $[0.5, 0.6]$. Due to the sampling procedure of BA, it is not possible to produce graph in this sparsity range, so we lowered the range sightly. We thus take SCs to be an SBM-like ensemble and avoid a repetitive experiment with randomly drawn SBM graphs.

Note that the edge density ranges define the performance of the most naive of predictors: all ones/zeros. In the RG/BA/SC, an all ones predictor achieves an average error of $44\% = 1 - (\text{average edge density})$. In the BAs, a naive all zeros predictor achieves $35\% = 1 - (\text{average edge density})$. This is useful to keep in mind when interpreting the results in Table 1.

**Pseudo-synthetics.** The 'pseudo-synthetic datasets' are those in which we diffuse synthetic signals over SCs from the HCP-YA dataset. This is an ideal setting to test the GDN models: we have weighted graphs to perform edge-regression on (the others are unweighted), while having $\boldsymbol{A}_O$'s that are true to our modeling assumptions. Note that SCs have a strong community-like structure, corresponding dominantly to the left and right hemispheres as well as subnetworks which have a high degree of connection, e.g. the Occipital Lobe which has 0.96 edge density - almost fully connected - while the full brain network has edge density of 0.56.

**Generation of the training set from synthetic diffused signals.** We carry out the following steps to generate the training set $\mathcal{T} := \{\boldsymbol{A}_O^{(i)}, \boldsymbol{A}_L^{(i)}\}_{i=1}^T$ for our (pseudo-)synthetic experiments in Section 5.1. The diffusion graph filter $\boldsymbol{H}(\cdot; \boldsymbol{h})$ is fixed across all observations in the training set. We let $K = 2$ and sample the filter coefficients $\boldsymbol{h} \in \mathbb{R}^3$ uniformly from the unit sphere. To generate each observation $\boldsymbol{A}_O^{(i)}$, we let $P = 50$ and first sample $\boldsymbol{w}_1^{(i)}, \ldots, \boldsymbol{w}_P^{(i)} \sim \mathcal{N}(\boldsymbol{0}, \boldsymbol{I}_N)$ (i.i.d. standard Normal random vectors). For the given filter $\boldsymbol{h}$, these white signals are diffused over $\boldsymbol{A}_L^{(i)}$ (using the graph models described above) to yield $\boldsymbol{x}_p^{(i)} = \boldsymbol{H}(\boldsymbol{A}_L^{(i)}; \boldsymbol{h})\boldsymbol{w}_p^{(i)}$, for each $p = 1, \ldots, P$. We finally form the sample covariance

$$\boldsymbol{A}_O^{(i)} = \boldsymbol{\Sigma}_x^{(i)} = \frac{1}{P}\sum_{p=1}^P \boldsymbol{x}_p^{(i)}\left(\boldsymbol{x}_p^{(i)}\right)^\top.$$

This process is repeated for each sample $i = 1, \ldots, T$ in the training set. To examine robustness to the realizations of $\boldsymbol{h}$, we repeat this data generation process three times (resampling the filter coefficients).

**Normalization.** Unless otherwise stated we divide $\boldsymbol{A}_O$ by its maximum eigenvalue before processing. $\boldsymbol{A}_L$ is used in its raw form when it is an unweighted graph. The SCs define weighted graphs and we scale all edge weights to be between 0 and 1 by dividing by 9.9 (an upper bound on the maximum edge weight in the HCP-YA dataset). Similar to (Shrivastava et al., 2020; Pu et al., 2021), we find GLAD to be very sensitive to the conditioning of the labels $\boldsymbol{A}_L$; see the Appendix A.7 for a full discussion on the normalization used in this case.

**Error and MSE of the baselines in Table 1.** The edge weights returned by the baselines can be very small/large in magnitude and perform poorly when used directly in the regression task. We thus also provide a learned scaling parameter, tuned during the hyperparameter search, which provides approximately an order of magnitude improvement in MSE in GLASSO and halved the MSE in Spectral Templates and Network Deconvolution. In link-prediction, we also tune a hard thresholding parameter on top of each method to clean up noisy outputs from the baseline, only if it improved their performance (it does). For complete details on the baseline implementation; see the Appendix A.7. Note also that the MSE reported in Table 1 is per-edge squared error - averaged over the test set, namely $\frac{1}{M}\frac{1}{|\mathcal{T}_{test}|}\sum_{i=1}^{\mathcal{T}_{test}}\|\Phi(\boldsymbol{A}_O^{(i)}) - \boldsymbol{A}_L^{(i)}\|_F^2$, where $M := N(N-1)$ is the number of edges in a fully connected graph (without self loops) with $N = 68$ nodes.

**Size generalization.** Something to note is that we do **not** tune the threshold (found during training on the small $N = 68$ graphs) on the larger graphs. We go straight from training to testing on the larger domains. Tuning the threshold using a validation set (of larger graphs) would represent an easier problem. The model at no point, or in any way, is introduced to the data in the larger size domains for any form of training/tuning.

We decide to use the covariance matrix in this experiment, as opposed to the sample covariance matrix, as our $\boldsymbol{A}_O$'s. This is for the simple reason that it would be difficult to control the SNR with respect to generalization error and would be secondary to the main thrust of the experiment. When run with number of signals $P$ proportional to graph size $N$, we see quite a similar trend, but due to time constraints, these results are not presented herein, but is apt for follow up work.

**HCP data.** HCP-YA provides up to 4 resting state fMRI scanning sessions for each subject, each lasting 15 minutes. We use the fMRI data which has been processed by the minimal processing pipeline (Van Essen et al., 2013). For every subject, after pre-processing the time series data to be zero mean, we concatenate all available time-series together and compute the sample covariance matrix (which is what as used are $\boldsymbol{A}_O$ in the brain data experiment 5.2).

Due to expected homogeneity in SCs across a healthy population, information about the SC in the test set could be leveraged from the SCs in the training set. In plainer terms, the average SC in the training set is

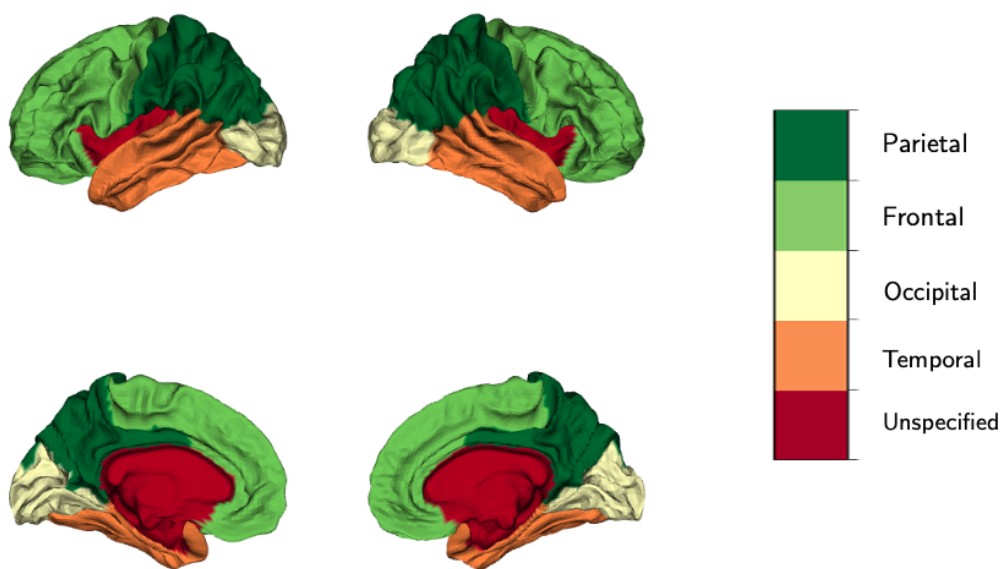

Figure 6: The four lobes in the brain cortex: parietal, frontal, occipital and temporal.

an effective predictor of SCs in the test set. In our experiments, we take random split of the 1063 SCs into 913/50/100 training/validation/test sets, and report how much we improve upon this predictor.

Raw data available from https://www.humanconnectome.org/study/hcp-young-adult/overview, and subject to the HCP Data Use Agreement. See here for a discussion on the risks of personal identifiability.

**Co-location and social networks among high school students.** The Thiers13 dataset (Génois & Barrat, 2018) followed students in a French high school in 2013 for 5 days, recording their interactions based on physical proximity (co-location) using wearable sensors as well as investigating social networks between students via survey. The co-location study traced 327 students, while a subset of such students filled out the surveys (156). We thus only consider the 156 students who have participated in both studies. The co-location data is a sequence of time intervals, each interval lists which students were within a physical proximity to one another in such interval. To construct the co-location network, we let vertices represent students and assign the weight on an edge between student $i$ and student $j$ to represent the (possibly 0) number of times over the 5 day period the two such students were in close proximity with one another. The social network is the Facebook graph between students: vertex $i$ and vertex $j$ have an unweighted edge between them in the social network if student $i$ and student $j$ have are friends in Facebook. Thus we now have a co-location network $\boldsymbol{A}_{O,\text{total}}$ and a social network $\boldsymbol{A}_{L,\text{total}}$. To construct a dataset of graph pairs $\mathcal{T} = \{\boldsymbol{A}_O^{(i)}, \boldsymbol{A}_L^{(i)}\}_{i=1}^{7000}$ we draw random subsets of $N = 120$ vertices (students). For each vertex subset, we construct a single graph pair by restricting the vertex set in each network to the sampled vertices, and removing all edges which attach to any node not in the subset. If either of the resulting graph pairs are not connected, the sample is not included.

We report performance of each model with an additional tuned threshold on the output, only if it increases performance. ND does not change the output meaningfully (and thus the performance is identical to simply thresholding). GLAD continually converged to local minimum with worse performance than simple thresholding. Due to the scaling issues - and resulting large computational costs - a subset of this dataset is used for ND (train/val/test of 500/100/200) and GLASSO/SpecTemp (train/val/test of 20/20/85). Increasing dataset sizes further did not seem to improve performance for any model.

## A.7 Baselines

In the broad context of network topology identification, recent DL-based latent graph inference approaches such as DGCNN (Wang et al., 2019), DGM (Kazi et al., 2023), NRI (Kipf et al., 2018), or PGN (Veličković

et al., 2020) have been shown effective in obtaining better task-driven representations of relational data for machine learning applications, or to learn interactions among coupled dynamical systems. However, because the proposed GDN layer does not operate over node features, none of these state-of-the-art methods are appropriate for tackling the novel network deconvolution problem we are dealing with here. Hence, for the numerical evaluation of GDNs we chose the most relevant baseline models that we outline in Section 5, and further describe in the sequel in no particular order.

We were as fair as possible in the comparison of GDNs to baseline models. All baselines were optimized to minimize generalization error, which is what is presented in Table 1. Many baseline methods aim to predict sparse graphs on their own, yet many fail to bring edge values fully to zero. We thus provide a threshold, tuned for generalization error using a validation set, on top of each method only if it improved the performance of the method in link-prediction. The edge weights returned by the baselines can be very small/large in magnitude and perform poorly when used directly in the the regression task. We thus also provide a scaling parameter, tuned during the hyperparameter search, which provides approximately an order of magnitude improvement in MSE for GLASSO and halved the MSE for Spectral Templates and Network Deconvolution.

**Unrolled PDS (L2G).** L2G (Pu et al., 2021) is an unrolling of the forward-backward-forward primal-dual splitting algorithm (PDS) introduced by (Kalofolias, 2016), which aims to recover sparse, connected graphs from observations of smooth signals w.r.t. the latent graph $\boldsymbol{A}_L$. More formally, consider the matrix $\boldsymbol{X} = [\boldsymbol{x}_1, \ldots, \boldsymbol{x}_P] \in \mathbb{R}^{N \times P}$ whose columns $\boldsymbol{x}_p$ are the observations in $\mathcal{X}$. The rows, denoted by $\bar{\boldsymbol{x}}_i^\top \in \mathbb{R}^{1 \times P}$ collect all $P$ measurements at vertex $i$. Define then the nodal Euclidean-distance matrix $\boldsymbol{E} \in \mathbb{R}_+^{N \times N}$, where $\boldsymbol{E}_{ij} := \|\bar{\boldsymbol{x}}_i^\top - \bar{\boldsymbol{x}}_j^\top\|_2^2, i, j \in \mathcal{V}$. Using these notions, and taking $\boldsymbol{A}_O = \boldsymbol{E}$ the signal smoothness measure of $\boldsymbol{X}$ over $\boldsymbol{A}$ can be equivalently written as

$$\sum_{p=1}^{P} \mathrm{TV}(\boldsymbol{x}_p) = \mathrm{Tr}\left(\boldsymbol{X}^\top \boldsymbol{L} \boldsymbol{X}\right) = \frac{1}{2}\|\boldsymbol{A} \odot \boldsymbol{A}_O\|_1 \tag{12}$$

where $\boldsymbol{L} = \mathrm{diag}(\boldsymbol{A}\boldsymbol{1}) - \boldsymbol{A}$ is the graph Laplacian and $\odot$ denotes the elementwise product (Kalofolias, 2016). We can use the link between signal smoothness and edge sparsity in (12) to pose a convex inverse problem

$$\hat{\boldsymbol{A}} \in \arg\max_{\boldsymbol{A} \in \mathcal{A}} \left\{ \|\boldsymbol{A} \odot \boldsymbol{A}_O\|_1 - \alpha\boldsymbol{1}^\top \log(\boldsymbol{A}\boldsymbol{1}) + \frac{\beta}{2}\|\boldsymbol{A}\|_F^2 \right\}, \tag{13}$$

where $\alpha, \beta$ are tunable regularization parameters and $\mathcal{A}$ denotes the convex set of allowable adjacency matrices (hollow diagonal, non-negative entries, symmetric).

We implemented L2G following the released code from the authors (Pu et al., 2021), which uses a depth of 20, a learning rate of 0.01, a learning rate decay of 0.95, and the ADAM optimizer. L2G uses intermediate outputs in the loss, and discount these intermediate losses using a loss-discounting factor $\gamma = 0.9$ which we also use in our implementation. We tried both with and without intermediate losses and did not find a significant difference. We do not use the additional Variational Autoencoder introduced in (Pu et al., 2021). Finally, for the link-prediction task, a threshold $\tau$ is chosen with a validation set to map the output to binary decision over edges in the same manner as used in GDNs and GLAD – see Appendix A.5.

**Graph Convolutional Network applied to degrees of $\boldsymbol{A}_O$ (GCN-D).** GCNs are simple graph neural network models introduced in (Kipf & Welling, 2017), which take node features $\boldsymbol{X} \in \mathbb{R}^{N \times C}$, a graph $\boldsymbol{A} \in \mathbb{R}^{N \times N}$, and uses weight matrices $\boldsymbol{W}_0, \boldsymbol{W}_1$ to produce latent node representation $\boldsymbol{Z} = \hat{\boldsymbol{A}}\mathrm{ReLU}(\hat{\boldsymbol{A}}\boldsymbol{X}\boldsymbol{W}_0)\boldsymbol{W}_1$. The graph $\hat{\boldsymbol{A}}$ is defined by the 'renormalization trick' $\hat{\boldsymbol{A}} := \tilde{\boldsymbol{D}}^{-\frac{1}{2}}\tilde{\boldsymbol{A}}\tilde{\boldsymbol{D}}^{-\frac{1}{2}}$, where $\tilde{\boldsymbol{A}} = \boldsymbol{A} + \boldsymbol{I}$ and $\tilde{\boldsymbol{D}} = \mathrm{diag}(\tilde{\boldsymbol{A}}\boldsymbol{1})$. In our experiments we take our graph to be the observed graph $\boldsymbol{A} = \boldsymbol{A}_O$, and our node signals to be the nodal degrees $\boldsymbol{x} = \boldsymbol{A}_O\boldsymbol{1} = \boldsymbol{d}$ ($C = 1$), producing the encoding model:

$$\boldsymbol{Z} = \hat{\boldsymbol{A}}_O\mathrm{ReLU}(\hat{\boldsymbol{A}}_O\boldsymbol{d}\boldsymbol{W}_0)\boldsymbol{W}_1 \tag{14}$$

To decode the latent representations into edge predictions we use a simple outer product decoder $\hat{\boldsymbol{A}}_L = \boldsymbol{Z}\boldsymbol{Z}^\top$. In (Kipf & Welling, 2016) a sigmoid is used on the output of the inner product to produce a probability, which is then fed into a binary cross entropy loss function. We found including this sigmoid did not produce

good graph estimates or stable training. So we remove it and directly use the outer product in our loss, which for link-prediction was a hinge loss and for regression it was a MSE loss.

We use as close to identical architectural and training setup to (Kipf & Welling, 2016) as possible: hidden/latent channel dimensions of 32/16, learning rate of 0.01, ADAM optimizer, training for 200 epochs (we use SGD with large batches - 20% of training data per batch - instead of full gradient descent as in (Kipf & Welling, 2016) to work with our GPU). We found that the standard ReLU did not allow rich gradient information to flow to the first layers weights, and so we replace it with a LeakyReLU with slope = 0.2 which increased performance. Additionally, the degree node signals $\boldsymbol{d} = \boldsymbol{A}_O \boldsymbol{1}$ result in very large values on many of our datasets, and so we normalize via $\boldsymbol{d}_{norm} = \frac{\boldsymbol{d}}{\|\boldsymbol{d}\|_2}$. A hyperparameter search was performed on the learning rate, but we could not find one which improved performance over the 0.01 used in (Kipf & Welling, 2016). We performed further hyperparameter searches on the channel widths (hidden, latent) $\in \{(8, 4), (16, 8), (32, 16), (64, 32)\}$, and found all worked similarly well, and so used $(32, 16)$ as in (Kipf & Welling, 2016). We also tested whether one-hot encoded features (included with and without the degree feature) could aid performance, but they did not.

**Graph Isomorphism Network applied to degrees of $\boldsymbol{A}_O$ (GIN-D).** GINs are graph neural network models introduced in (Xu et al., 2019), which take node features $\boldsymbol{X} \in \mathbb{R}^{N \times C}$, a graph $\boldsymbol{A} \in \mathbb{R}^{N \times N}$ and updates node representations as:

$$\boldsymbol{z}_v^{(k)} = \text{MLP}^{(k)}\big((1 + \epsilon^{(k)})\boldsymbol{z}_v^{(k-1)} + \sum_{u \in \mathcal{N}(v)} \boldsymbol{z}_u^{(k-1)}\big) \tag{15}$$

In the first layer, the node representation are set to the input node features. Here the input node signals to be $\boldsymbol{x} = \boldsymbol{A}_O \boldsymbol{1} = \boldsymbol{d}$ (C=1). Note our graphs are weighted, and so in the sum we also use the edge weights occurring in the graph. Self-loops can occur, for example as variances on the diagonal in constructed covariance matrices, but are not included in the neighborhood of a node. Similar to the GCN-D baseline, to decode the latent representations into edge predictions we use an outer product decoder $\hat{\boldsymbol{A}}_L = \boldsymbol{Z}\boldsymbol{Z}^\top$, where $\boldsymbol{Z}$ is the matrix of latent node representations $\boldsymbol{z}_v^{(D)}$ in the final layer. For link prediction we additionally feed this through a sigmoid and then a binary cross entropy loss, for edge regression we feed this straight to an MSE loss.

We use as close to identical architectural and training setup to (Xu et al., 2019) as possible: $D = 5$ GIN layers (including the input layer) are applied, all MLPs have 2 layers, hidden channel dimensions of 64, the $\epsilon^{(k)}$ in all layers are not learned and set to 0 - denoted GIN-0 in (Xu et al., 2019), batch normalization is applied on each hidden layer, a learning rate of 0.01, ADAM optimizer, decay learning rate by 0.5 every 50 epochs, and training was performed for 350 epochs and stopped early after 40 epochs if no improvement in loss was found. Additionally, the degree node signals $\boldsymbol{d} = \boldsymbol{A}_O \boldsymbol{1}$ result in very large values on many of our datasets, and so we normalize via $\boldsymbol{d}_{norm} = \frac{\boldsymbol{d}}{\|\boldsymbol{d}\|_2}$.

**Hard Thresholding (Threshold).** The hard thresholding model consists of a single parameter $\tau$, and generates graph predictions as follows

$$\hat{\boldsymbol{A}}_L = \mathbb{I}\big\{|\boldsymbol{A}_O| \succeq \tau \boldsymbol{1}\boldsymbol{1}^\top\big\},$$

where $\mathbb{I}\{\cdot\}$ is an indicator function, and $\succeq$ denotes entry-wise inequality. For the synthetic experiments carried out in Section 5.1 to learn the structure of signals generated via network diffusion, $\boldsymbol{A}_O$ is either a covariance matrix or a correlation matrix. We tried both choices in our experiments, and reported the one that performed best in Table 1.

**Graphical Lasso (GLASSO).** GLASSO is an approach for Gaussian graphical model selection (Yuan & Lin, 2007; Friedman et al., 2008). In the context of the first application domain in Section 3, we will henceforth assume a zero-mean graph signal $\boldsymbol{x} \sim \mathcal{N}(\boldsymbol{0}, \boldsymbol{\Sigma}_x)$. The goal is to estimate (conditional independence) graph structure encoded in the entries of the precision matrix $\boldsymbol{\Theta}_x = \boldsymbol{\Sigma}_x^{-1}$. To this end, given an empirical covariance matrix $\boldsymbol{A}_O := \hat{\boldsymbol{\Sigma}}_x$ estimated from observed signal realizations, GLASSO regularizes the maximum-likelihood estimator of $\boldsymbol{\Theta}_x$ with the sparsity-promoting $\ell_1$ norm, yielding the convex problem

$$\hat{\boldsymbol{\Theta}} \in \arg\max_{\boldsymbol{\Theta} \succeq \boldsymbol{0}} \big\{\log\det\boldsymbol{\Theta} - \text{trace}(\hat{\boldsymbol{\Sigma}}_x\boldsymbol{\Theta}) - \alpha\|\boldsymbol{\Theta}\|_1\big\}. \tag{16}$$

We found that taking the entry-wise absolute value of the GLASSO estimator improved its performance, and so we include that in the model before passing it through a hard-thresholding operator

$$\hat{\boldsymbol{A}}_L = \mathbb{I}\left\{ |\hat{\boldsymbol{\Theta}}| \succeq \tau \mathbf{1}\mathbf{1}^\top \right\}$$

One has to tune the hyperparameters $\alpha$ and $\tau$ for link-prediction (and a third, the scaling parameter described below, for edge-weight regression).

We used the sklearn GLASSO implementation found in: `https://scikit-learn.org/stable/modules/generated/sklearn.covariance.graphical_lasso.html`

It is important to note that we do **not** use the typical cross-validation procedure seen with GLASSO. Typically, GLASSO is used in unsupervised applications with only one graph being predicted from $P$ observations. In our application, we are predicting many graphs, *each* with $P$ observations. Thus the typical procedure of choosing $\alpha$ using the log-likelihood [the non-regularized part of the GLASSO objective in (16)] *over splits of the observed signals, not splits of the training set*, results in worse performance (and a different $\alpha$ for each graph). This is not surprising: exposing the training procedure to labeled data allows it to optimize for generalization. We are judging the models on their ability to generalize to unseen graphs, and thus the typical procedure would provide an unfair advantage to our model. While we tried both sample covariance and sample correlation matrices as $\boldsymbol{A}_O$, we found that we needed the normalization that the sample correlation provides, along with an additional scaling by the maximum magnitude eigenvalue, in order to achieve numerical stability. GLASSO can take a good amount of time to run, and so we limited the validaiton and test set sizes to an even 100/100 split. With only 2 to 3 hyperparameters to tune, we found this was sufficient (no significant differences between validation and test performance in all runs, and when testing on larger graph sizes, no difference in generalization performance).

**Unrolled AM on GLASSO Objective (GLAD).** GLAD is an unrolling of an alternating minimization (AM) iterative procedure to form the GLASSO estimator (16); refer to (Shrivastava et al., 2020) for full details. GLAD inherits sensitivity to the conditioning of the labels from the constraint that $\boldsymbol{\Theta}$ be positive semi-definite in (16). In the experiments presented in (Shrivastava et al., 2020), precision matrices are sampled directly to be used as labels, and their minimum eigenvalue is set to 1 by adding an appropriately scaled identity matrix (diagonal loading). This minimum eigenvalue correction was attempted on the raw adjacency matrices $\boldsymbol{A}_L$ but did not result in convergent training on any data domain in our experiments. The best results (reported in Table 1) were found corresponding to the minimum eigenvalue corrected Laplacian matrix representation of the latent graph, i.e., $\boldsymbol{L}_L := \mathrm{diag}(\boldsymbol{A}_L \mathbf{1}) - \boldsymbol{A}_L + \boldsymbol{I}$. The Laplacian $\boldsymbol{L}_L$ – referred to as $\boldsymbol{\Theta}^*$ in (Shrivastava et al., 2020) – is used *only in the the loss function* to make training converge. The reported metrics in Table 1 disregard the diagonal elements and compute error and MSE based on the off-diagonal entries, thus ensuring an equivalent comparison with all other methods. Finally, for the link-prediction task, a threshold $\tau$ is chosen with a validation set to map the output to binary decision over edges in the same manner as used in GDNs; see also Appendix A.5.

We use the GLAD configuration as presented in (Shrivastava et al., 2020): $L = 30$ layers, ADAM optimizer with learning rate lr $= 0.1$ and $\beta_1 = 0.9$, $\beta_2 = 0.999$, the $\rho_{nn}$ has 4 hidden layers, and $\Lambda_{nn}$ has 2 hidden layers. An intermediate loss discount factor $\gamma = 0.8$ worked reasonably well in the domains tested. *ER* graphs caused GLAD to converge to a trivial all zeros solution when $D = 30$. We increased the depth to $D = 50$ and performed a search over the learning rate and minimum eigenvalue $m_e$ (previously set to 1), resulting in a GLAD model ($D = 50$, $lr = .05$, $m_e = 100$, all other identical) that decreased error (by ~1%) in the link-prediction task relative to the naive all zero output. MSE got worse in this configuration relative to the previous. In Table 1 we report respective metrics from their best configurations: the error with the deeper configuration and $MSE$ with the shallower configuration.

When training on the SCs in Table 1, we incorporate prior information in GLAD by setting the prior to $\boldsymbol{A}[0] = \mathrm{diag}(\bar{\boldsymbol{A}}\mathbf{1}) - \bar{\boldsymbol{A}} + \boldsymbol{I}$, where $\bar{\boldsymbol{A}}$ is the adjacency matrix of the edgewise mean graph over all SCs in the training split. In other words we use the minimum eigenvalue corrected Laplacian matrix (as above) representation of $\bar{\boldsymbol{A}}$. GDNs simply use $\bar{\boldsymbol{A}}$ as discussed in Section 5. We also tried the default setup, where the prior is $\boldsymbol{A}[0] = (\boldsymbol{A}_O + \boldsymbol{I})^{-1}$, but found inferior performance.

GLAD in general has an $\mathcal{O}(DN^3)$ run time complexity, where $D$ is the depth of GLAD, in both the forward and the backward pass, due to the computation of the matrix square root and its corresponding gradient. These can be approximated using e.g. Denman-Beavers iterations or Newton-Schulz iterations, which reduce the time complexity to $\mathcal{O}(DTN^2)$, where $T$ is the number of iterations run. These iterative methods still lack guarantees of convergence in general. We used such approximations without noticing reductions in performance to make the experiments run in reasonable amounts of time. Even with such approximations GLAD has significant memory and runtime limitations. This results from the large $D$, moderate sized $T$ (required for reasonable/stable approximations), as well as the use of shared intermediate MLPs - $\rho_{nn}$ and $\Lambda_{nn}$ - the former of which is called $N^2$ times (one for each entry in $\boldsymbol{\Theta}$) *per layer*. This forces batch sizes smaller than what would be desired and limits applicability on larger graph sizes. Note that in contrast the dominant term in the memory & runtime complexity of GDN layers are the 2 (typically sparse) matrix multiplications, making training and deployment on larger graph sizes realizable; see Figure 3.

**Network Deconvolution (ND).** The ND approach is "a general method for inferring direct effects from an observed correlation matrix containing both direct and indirect effects" (Feizi et al., 2013). ND follows three steps: linear scaling to ensure all eigenvalues $\lambda_i$ of $\boldsymbol{A}_O$ fall in the interval $\lambda_i \in [-1, 1]$, eigen-decomposition of the scaled $\boldsymbol{A}_O = \boldsymbol{V}\text{diag}(\boldsymbol{\lambda})\boldsymbol{V}^{-1}$, and deconvolution by applying $f(\lambda_i) = \frac{\lambda_i}{1+\lambda_i}$ to all eigenvalues. We then construct our prediction as $\hat{\boldsymbol{A}}_L := \boldsymbol{V}\text{diag}(f(\boldsymbol{\lambda}))\boldsymbol{V}^{-1}$. In (Feizi et al., 2013), it is recommended a Pearson correlation matrix be constructed, which we followed. We applied an extra hard thresholding on the output, tuned for best generalization error, to further increase performance. For each result shown 500 graphs were used in the hyperparameter search and 500 were used for testing.

**Spectral Templates (SpecTemp).** The SpecTemp method consists of a two-step process whereby one: (i) first leverages the model (1) to estimate the graph eigenvectors $\boldsymbol{V}$ from those of $\boldsymbol{A}_O$ (the eigenvectors of $\boldsymbol{A}_L$ and $\boldsymbol{A}_O$ coincide); and (ii) combine $\boldsymbol{V}$ with a priori information about $G$ (here sparsity) and feasibility constraints on $\mathcal{A}$ to obtain the optimal eigenvalues $\boldsymbol{\lambda}$ of $\boldsymbol{A}_L = \boldsymbol{V}\text{diag}(\boldsymbol{\lambda})\boldsymbol{V}^\top$.

The second step entails solving the convex optimization problem

$$\boldsymbol{A}^*(\epsilon) := \underset{\{\boldsymbol{A}, \bar{\boldsymbol{\lambda}}\}}{\text{argmin}} \|\boldsymbol{A}\|_1, \quad \text{s. to } \|\boldsymbol{A} - \boldsymbol{V}\text{diag}(\bar{\boldsymbol{\lambda}})\boldsymbol{V}^\top\|_2^2 < \epsilon, \ \boldsymbol{A}\mathbf{1} \succeq \mathbf{1}, \ \boldsymbol{A} \in \mathcal{A}. \quad (17)$$

We first perform a binary search on $\epsilon \in \mathbb{R}_+$ over the interval $[0, 2]$ to find $\epsilon_{min}$, which is the smallest value which allows a feasible solution to (17). With $\epsilon_{\min}$ in hand, we now run an iteratively ($t$ henceforth denotes iterations) re-weighted $\ell_1$-norm minimization problem with the aim of further pushing small edge weights to 0 (thus refining the graph estimate). Defining the weight matrix $\boldsymbol{W}_t := \frac{\gamma\mathbf{11}^\top}{|\boldsymbol{A}_{t-1}^*|+\delta\mathbf{11}^\top} \in \mathbb{R}_+^{N \times N}$ where $\gamma, \delta \in \mathbb{R}_+$ are appropriately chosen positive constants and $\boldsymbol{A}_0^* := \boldsymbol{A}^*(\epsilon_{min})$, we solve a sequence $t = 1, \ldots, T$ of weighted $\ell_1$-norm minimization problems

$$\boldsymbol{A}_t^* := \underset{\{\boldsymbol{A}, \bar{\boldsymbol{\lambda}}\}}{\text{argmin}} \|\boldsymbol{W}_t \odot \boldsymbol{A}\|_1, \quad \text{s. to } \|\boldsymbol{A} - \boldsymbol{V}\text{diag}(\bar{\boldsymbol{\lambda}})\boldsymbol{V}^\top\|_2^2 < \epsilon_{min}, \ \boldsymbol{A}\mathbf{1} \succeq \mathbf{1}, \ \boldsymbol{A} \in \mathcal{A}. \quad (18)$$

If any of these problems is infeasible, then $\boldsymbol{A}_I^*$ is returned where $I \in \{0, 1, \ldots, T\}$ is the last successfully obtained solution.

Finally, a threshold $\tau$ is chosen with a validation set to map the output to binary decision over edges, namely

$$\hat{\boldsymbol{A}}_L = \mathbb{I}\left\{|\boldsymbol{A}_I^*| \succeq \tau\mathbf{11}^\top\right\},$$

We solve the aforementioned optimization problems using MOSEK solvers in CVXPY. Solving such convex optimization problems can very computationally expensive/slow, and because there are only two hyperparameters, 100 graphs were used in the validation set, and 100 in the test set.

**Least Squares plus Non-convex Optimization (LSOpt).** LSOpt consists of first estimating the polynomial coefficients in (1) via least-squares (LS) regression, and then using the found coefficients in the optimization of (3) to recover the graph $\boldsymbol{A}_L$. Due to the collinearity between higher order matrix powers, ridge regression can be used in the estimation of the polynomial coefficients. If the true order $K$ of the polynomial is not known ahead of time, one can allow $\hat{\boldsymbol{h}} \in \mathbb{R}^N$ and add an additional $\ell_1$-norm regularization

Table 3: Runtime and memory usage analysis. GDN, L2G, and GLAD report time (s) for a forward and backward pass on a single sample (batch size $= 32$), as well as peak GPU memory consumption, using a T4 GPU on a $g4dn$.xlarge AWS instance. SpecTemp reports inference time (s) for a single sample, and peak memory consumption, using a 96 Core x86 Processor on a c5.metal AWS instance. Results are taken after 40 epochs/samples to allow the running process to stabilize.

| MODELS | TIME (S) PER SAMPLE | PEAK MEMORY USAGE (GB) |
|---|---|---|
| GDN-1 | $8.7\text{E-}4_{\pm 1\text{E-}2}$ | 1.28 |
| GDN-8 | $3.1\text{E-}3_{\pm 7\text{E-}3}$ | 12.84 |
| L2G | $2.2\text{E-}3_{\pm 3\text{E-}3}$ | 1.11 |
| GLAD | $6.7\text{E-}3_{\pm 8\text{E-}3}$ | 11.73 |
| SPECTEMP | $11.3\text{E-}0_{\pm 5\text{E-}1}$ | 3.84 |

for the sake of model-order selection. With $\hat{\boldsymbol{h}}$ in hand, we optimize (3), a non-convex problem due to the higher-order powers of $\boldsymbol{A}$, using ADAM with a learning rate of 0.01 and a validation set of size 50 to tune $\lambda$.

We start with the most favorable setup, using the true $\boldsymbol{h}$ in the optimization of (3), skipping the LS estimation step. Even in such a favorable setup the results were not in general competitive with GDN predictions, and so instead of further degrading performance by estimating $\hat{\boldsymbol{h}}$, we report these optimistic estimates of generalization error.

### A.8 Additional Numerical Experiments

**Runtime and memory usage.** In Section 5.1 we discuss the runtime complexity of GDNs and L2G (quadratic) as well as the runtime complexity of the cubic baselines (GLAD, GLASSO, ND, SpecTemp). We further discuss in Section A.7 the approximations made to the matrix square root in GLAD to reduce its runtime complexity to be quadratic. Here we explore the actual run time (s) per sample and peak memory usage (for the full batch) for a handful of models taken from the runs on the $N = 68$ RG graphs in Table 1. Note that GDNs, L2G, and GLAD are DL models trained with stochastic gradient descent, whereas SpecTemp is the solution to a sequence of optimization problems. We thus report the time to run the forward and backward pass over a batch of size 32 (we normalize by batch size to get a per sample time) for the former and time for the full inference over a single sample in the latter. GDN-1 refers to a GDN model with $D = 8$ and $C = 1$, i.e. no MIMO channels, while GDN-8 refers to a GDN model $D = 8$ and $C = 8$ - the same model used for the experiments in Table 1. GLAD, L2G, and SpecTemp also follow the same setup used in Table 1 - see Section A.7 for details. The fundamental operations in GDNs are batched matrix multiplications and additions - both of which are highly parallelizable and, importantly, have optimized implementations available to us. This is particularly true for the case of sparse batches of matrices (a setting where we can approach a *linear* runtime complexity). While we were able to reduce GLAD's runtime complexity to be quadratic, it still requires multiple internal MLPs and a deeper unrolled network for good performance. SpecTemp relies on the full eigendecomposition and solving multiple internal convex optimization problems each time it performs inference for a sample - posing issues for real time applications or large networks.

The results in Table 3 show GDNs have the fastest runtimes, with GDN$-1$'s being $\sim 3x$ ($8x$) faster and using $\sim 10x$ ($10x$) less peak GPU memory than GDN$-8$ (GLAD). GDN$-8$ is about twice as fast as GLAD, with similar peak GPU memory usage. SpecTemp is orders of magnitude slower - and uses significantly more memory per sample - owing to the cubic operations it contains.

**GDN hyperparameter search.** To explore the effect of the dominant hyperparameters in GDNs, we perform a simple grid search over Depth (D) $\in \{3, 8, 20\}$ and Channels per Layer (C) $\in \{1, 3, 8, 15\}$ on RG graphs used in Table 1 (train/val/test of 913/500/500). We did not include learning rate in the search as we found around the value of .01 there was no major effect on performance. As can be seen in Table 4, for each fixed depth, there is a monotonic improvement in performance (decrease in error) with an increasing number of channels. This occurs until our GPU runs out of memory (OOM). For each fixed channel width, there seems to be an improvement in performance when depth is increased from $D = 3$ to $D = 8$, then a proceeding

Table 4: Hyperparameter Search over Depth (D) and Channels per Layer (C) of a GDN network. Reported are the mean and standard error of the test error (%) on the link prediction task. OOM indicates 'Out of Memory'.

| D $\backslash$ C | 1 | 3 | 8 | 15 |
|---|---|---|---|---|
| 3 | 7.0E-2$_{\pm1\text{E-}3}$ | 4.8E-2$_{\pm7\text{E-}4}$ | 3.5E-2$_{\pm5\text{E-}4}$ | OOM |
| 8 | 5.1E-2$_{\pm9\text{E-}4}$ | 2.9E-2$_{\pm3\text{E-}4}$ | 2.2E-2$_{\pm3\text{E-}4}$ | OOM |
| 20 | 5.1E-2$_{\pm7\text{E-}3}$ | 4.0E-2$_{\pm6\text{E-}4}$ | OOM | OOM |

degradation (or lack of improvement) in performance when $D = 20$. We thus choose $C = 8$ and $D = 8$ to get the best performance, while staying within the memory constraints of our hardware. These results would suggest further improvements in performance are readily attainable with more compute resources.

The full list of hyperparameters are as follows: Learning rate, depth $D$, channel widths $C$, normalization of input $\boldsymbol{A}_O$, normalization of intermediate outputs $\boldsymbol{A}[k]$. In searching over these hyperparameters, values for the learning rate were $\in \{1e - 6, 1e - 4, 1e - 2, 1e - 1, 1\}$, $D \in \{3, 8, 20\}$, $C \in \{1, 3, 8, 15\}$, normalization for $\boldsymbol{A}_O, \boldsymbol{A}[k] \in \{$maximum eigenvalue, frobenius norm, maximum absolute value norm, 99th percentile norm$\}$. Optimal values are learning rate $= 1e - 2, D = 8, C = 8$, and $\boldsymbol{A}_O, \boldsymbol{A}[k]$ normalization's of maximum eigenvalue, and maximum absolute value norm, respectively.

The procedure used to find such hyperparameter were as follows. Initially fix depth $D = 10$ with channel width $C = 1$. This was inspired by unrollings in other domains which tend to use in the 10's of layers with good performance, while not approaching the super deep networks which can have gradient vanishing problems. A hyperparameter search over appropriate normalization $\boldsymbol{A}_O \in \{$maximum eigenvalue, frobenium norm, maximum absolute value norm, 99th percentile norm$\}$ was performed first. Maximum eigenvalue normalization was found to perform best - and resulted in the most stable training. Next, a search over appropriate normalization for intermediate outputs $\boldsymbol{A}[k]$ was performed over the same normalizations. We found both maximum eigenvalue and maximum absolute value norm to perform similarly, but maximum eigenvalue normalization has higher computational cost and maximum absolute value norm projected the edges onto the interval $[-1, 1]$ which was convenient for our edge prediction applications. Next a search was performed over learning rate $\in \{1e - 6, 1e - 4, 1e - 2, 1e - 1, 1\}$, $D \in \{3, 8, 20\}$ and $C \in \{1, 3, 8, 15\}$. This search grid is displayed in Table 4 in A.8, with the learning rate dimension omitted for ease of interpretation. All searches were performed over RG graphs from Table 1.

### A.9 Source Code and Configuration

Code is available on github at `https://github.com/maxwass/pyGSL`. We rely on PyTorch (Paszke et al., 2019) (BSD license) heavily and use Conda (Anaconda, 2020) (BSD license) to make our system as hardware-independent as possible. GPUs were used to train the larger GDNs, which are available to use for free on Google Colab. Development was done on Mac and Linux (Ubuntu) systems, and not tested on Windows machines. The processed brain data was too large to include with the code, but is available upon request.

