# OpenReview forum: "Learning Graph Structure from Convolutional Mixtures"
_TMLR — Accepted by TMLR_

### Review · Reviewer_XTUC · 2023-01-31

**Summary Of Contributions:**

This paper systematically studies the problem of how to infer the graph structure from (partially) observed graphs as an optimization problem, which they denote as a network inverse/deconvolution problem. To solve this problem, the authors propose a neural network-based solution, GDN, from the principle of algorithm unrolling, and use truncated proximal projection. GDNs are differentiable with respect to the neural network parameters and the input graphs and are proven to have a node permutation equivariance property. Empirical evaluations on both synthetic and two real-world datasets, i.e., neuroimaging and social network datasets, demonstrate the effectiveness of GDNs on both link prediction and edge-weight regression tasks.

**Audience:**

Yes

**Claims And Evidence:**

Yes

**Requested Changes:**

My concerns are mainly from the mighty misleading name deconvolution, the discussion on how node features affect the performance of GDNs, and the runtime and memory usage comparison with L2G. Please kindly refer to Section Strengths and Weaknesses for detailed suggestions.

**Strengths And Weaknesses:**

Strengths:
* The investigation of the network inverse/deconvolution problem is interesting, which is claimed to have broad applications across various domains, ranging from inferring protein contact structure to epidemiology.
* The discussion on the problem background is comprehensive, which is very helpful for justifying the applicability of GDNs. The two discussed properties of GDNs, i.e., symmetry preserving and node permutation equivariance, are closely connected with the family of GNNs.
* The paper is generally well-written and almost clear everywhere. I enjoyed reading through this manuscript.

Weaknesses:
* The outline of the manuscript is a bit weird given the related work is embedded in the Introduction Section, but this does not really hurt the understandability.

* One concern from mine is the naming of network deconvolution, which is not very clearly defined in my opinion. In other literature such as [1], the process of deconvolution on graphs is explicitly denoted as recovering the original node features/signals given smoothed node representations obtained by a GNN. This deconvolution on the node features/signals is different from the notation of network deconvolution in this manuscript, which is indeed a network reverse problem and focuses on the graph structures. I suggest the authors define the task of network deconvolution more explicitly, with possible examples from the literature. The discussion on this difference would help audiences not be confused about the task dealt with in this paper.

* Since the architecture of GDNs does not involve the node features, could the authors provide some insights on how the node features on graphs might affect the performance of GDNs? For example, if there exist two graphs with different node features but the same network structure $\boldsymbol A_{O}$
reveal different latent graph $\boldsymbol A_{L}$ , could GDNs distinguish the two graphs during either the training or inference stage?

* For the empirical evaluation, the discussion on runtime and memory usage is favored. I am curious about why L2G[2] is not included in either complexity analysis or numerical comparison since L2G and Glad are the most related attempts to GDNs.

[1] Deconvolutional Networks on Graph Data, NeurIPS 2021

[2] Learning to learn graph topologies, NeurIPS 2021

---

> ### Author Response · Authors · 2023-03-13
> **Authors’ response to Reviewer XTUC**
>
> Thanks for your time and effort in reviewing our manuscript, as well as for finding the problem setting interesting and its applicability well justified. We are glad to hear you enjoyed reading the manuscript. Moreover, we appreciate your valuable suggestions to improve the paper’s presentation. Point-by-point responses to your comments and associated requests for changes follow. We strive to improve our paper and will be happy to continue the discussion if any outstanding issues remain.
>
> &nbsp;
>
> **On the name ‘Graph Deconvolution Network’.**
>
> The name is inspired from the graph signal processing literature as well as machine learning on graphs, where a ‘graph convolutional’ operator is defined as a polynomial of a matrix representation of the graph. Recall the data model (1) is $\mathbf{A}_O=h_0\mathbf{I} + h_1\mathbf{A}_L +  \ldots + h_K\mathbf{A}_L^K$,
>
> which takes the form of a ‘graph convolution’ or ‘graph convolutional filter’; see e.g.,
>
> - F. Gama et al, “Graphs, convolutions, and neural networks: From graph filters to graph neural networks,” IEEE Signal Process. Mag., 2020 https://arxiv.org/abs/2003.03777
> - A. Ortega et al, “Graph signal processing: Overview, challenges and applications,” Proc. IEEE, 2018 https://arxiv.org/abs/1712.00468
> - E. Isufi et al, “Graph filters for signal processing and machine learning on graphs”, IEEE Trans. Signal Process. 2022 https://arxiv.org/abs/2211.08854
>
> In Section 2 - Problem Formulation we state our goal: to recover sparse $\mathbf{A}_L$ from its graph convolutional mixture $\mathbf{A}_O$ as per (1) and in a supervised setting, thus motivating the use of ‘graph deconvolution’. As we discuss immediately after (1), one can think of $\mathbf{A}_O$ as a graph containing spurious, indirect connections generated by the terms including higher-order powers of latent graph $\mathbf{A}_L$ – the graph of fundamental relationships we wish to recover. This is exactly the same general problem description as in
>
> - S. Feizi et al, “Network deconvolution as a general method to distinguish direct dependencies in networks,” Nat. Biotechnol, 2013 https://www.nature.com/articles/nbt.2635,
>
> where arguably the term “network deconvolution” was coined for the first time (see also our related discussion in Section 3). We attempt to clarify this terminology in a few locations, including the Abstract “​​In this paper, we postulate a graph convolutional relationship between the observed and latent graphs, and formulate the graph structure learning task as a network inverse (deconvolution) problem.” If you feel a dedicated remark in e.g., Section 2 or the Appendix would help to further clarify terminology, we would be glad to include it in the revised paper we are preparing.
>
> Thanks for bringing up the related work in
> - J. Li et al, “Deconvolutional networks on graph data”, NeurIPS, 2021 https://arxiv.org/abs/2110.15528
>
> which investigates a different inverse problem of recovering input graph signals (nodal features) from representations obtained via a graph convolutional network (GCN). Notice that GCNs rely on graph convolutional filters such as those in (1), but with $K=1$ as we point out just before (5).  In the revised manuscript, we will cite this paper and clarify its different take on deconvolution over graphs under Remark (Graph convolutional data model in context) in Section 3.
>
> &nbsp;
>
> **GDN and node features.**
>
>
> This point is well taken. In the current formulation of GDNs, node features are not directly used and thus the learned network function would map any equivalent $\mathbf{A}_O$’s to identical $\hat{\mathbf{A}}_L$’s. We touch upon the important issue of uniqueness and identifiability of our model (which by definition does not involve nodal features) in Appendix A.1 - Model identifiability. But granted, if one has access to other features along with $\mathbf{A}_O$, e.g. age and gender in the neuroimaging domain, then they can be included in the final layer before prediction to resolve potential ambiguities. While this enhancement to the GDN model is beyond the scope of this already fully-packed paper, in Appendix A.4 - Incorporating Node Features we discuss one way to incorporate nodal features into GDNs with standard deep learning methods.

---

> > ### Author Response · Authors · 2023-03-13
> > **Authors’ response to Reviewer XTUC (Part 2)**
> >
> > **On complexity comparisons.**
> >
> >
> > Thanks for raising this point. L2G’s runtime and memory complexity are also quadratic in $N$, which we now explicitly state in Section 5.1 - Scaling and size generalization: Deploying on larger graphs. In the revised paper, we have included a line in Table 3 (A.8 - Additional Numerical Experiments) reporting L2G’s runtime and memory usage.
> >
> > Our approximate implementation of GLAD also incurs quadratic complexity; but even with such approximations GLAD has significant memory and runtime limitations as we discuss in Appendix A.7 - Baselines. We have mostly focused on comparisons with GLAD  because the former was the most performant in our experiments; see Table 1. We unintentionally overlooked stating the quadratic scaling of L2G, now clearly spelled out in the revised paper.
> >
> > &nbsp;
> >
> > **On the placement of the Related Work section.**
> >
> >
> > We opted to discuss the relevant work in the Introduction so that our novel technical contributions (summarized immediately after) can be better positioned and appreciated in context. In our experience that is the preferred practice, but we will be happy to place the section elsewhere if you believe it will improve the clarity of presentation.
> >
> > &nbsp;
> >
> > Thanks again for your review.

---

> > > ### Comment · Reviewer_XTUC · 2023-03-23
> > > **Thank you for the response**
> > >
> > > Thank you for the detailed and thoughtful feedback. I have also read the comments from other reviewers as well as the corresponding replies. My concerns are mostly addressed. Hopefully, the authors could include these points as well as the discussion in the updated version.

---

> > > > ### Author Response · Authors · 2023-03-24
> > > > **Revisions implemented in the updated version of the draft**
> > > >
> > > > Thanks again for your feedback as well as for engaging in the discussion. We are glad to hear that your concerns have been addressed.
> > > >
> > > > Indeed, last week we uploaded a revised version of the paper which incorporates the requested changes and discussions.

---

### Review · Reviewer_AruC · 2023-02-17

**Summary Of Contributions:**

This paper considered supervised graph learning problems where the observed graph is represented by a polynomial of the true graph structure.
For this problem, the paper proposed Graph Deconvolution Network (GDN). This model is obtained by linearizing a single step of optimization by proximal gradient descent of an L1-regularized loss function.
Several architectural improvements (prior optimization, multi-input multi-output filter, and decoupled layer paramters) were also proposed to improve the accuracy.
The proposed method was applied to the tasks of link prediction and edge weight regression on synthetic data and two types of real data (neuroimaging data and social network data), and the usefulness of the proposed method was verified by conducting an ablation study.

**Audience:**

Yes

**Broader Impact Concerns:**

This paper does not have major concerns about broader impacts.

**Claims And Evidence:**

Yes

**Requested Changes:**

[R1] I would like to clarify what the authors meant by "the novel supervised setting for topology identification" in the Summary of Main Contributions was the content of the Remark in Section 3. In other words, this paper considered the problem of estimating the true graph from the observed graph, which included the problem of estimating the graphical model from an empirical variance-covariance matrix (or precision matrix).

[R2] That paper claimed that "the GDN learns the distribution of the graph" (e.g., in the abstract). I would like to clarify that this claim meant that the GDN learns $P(y|x)$, where $x$ is the corrupted graph and $y$ is the unknown true graph structure.

[R3] Similar terms have been used in different contexts, which may confuse readers who are not familiar with this area. Specifically, I would like to clarify whether "graph structure learning" (abstract), "network topology inference" (P2), "latent graph learning" (P2), "topology identification" (P2), and "graph recovery" (abstract) all referred to the same task or different concepts.

[R4] P10, Section 5.1: I would like to clarify how the initial value of the block diagonal initialization was determined.

[R5] Section 5.1: I would like to clarify how to generate $A^{(i)}\_{O}$ (i.e. $\hat{\Sigma}\_{x}^{(i)}$) from $A^{(i)}\_{L}$. Is the following correct?

$w^{(i)}_1, \ldots, w^{(i)}_P \sim N(0, I)$

$x^{(i)}_p = H(A^{(i)}L)w^{(i)}_p$

$\hat{\Sigma}\_{x}^{(i)} = \frac{1}{P} \sum\_{p=1}^P x^{(i)}\_p x^{(i)\top}\_p$

[R6] Some citations were not formatted correctly. For example, Bronstein et al. (2017) and Ortega et al. (2018) in Section 1 should be replaced with (Bronstein et al. ) and (Ortega et al., 2018), respectively (i.e., use citep).

[R7] I think we can simplify the notation $\bm{H}(\bm{A}; \bm{h})$ to $\bm{H}(\bm{A})$ because the second argument $\bm{h}$ has not changed in this paper. We can interpret it as an assignment of $A$ to $\bm{H}(x) = \sum_{k=0}^K h_k x^k$.

[R8] The E notation in Table 1 and Table 2 was different.

[R9] References: Some references did not give the source (e.g., Pu et al. (2021), Kipf & Welling (2016)).

**Strengths And Weaknesses:**

**Strengths.**

- [St1] The proposed method is derived by a principle idea of approximating the optimization process ([So1]).
- [St2] The proposed method is generalized to graphs whose size is different from those in the training data set ([So3, N2]).
- [St3] In the experiments, the proposed method is compared with many baselines, and the experimental conditions of the baselines are taken into account so that the evaluation is fair. ([So4])

**Weaknesses**

- [W1] Writing of related work has room for improvement ([C1, C2])
- [W2] No comparison with existing algorithms for solving link prediction tasks ([So5])
- [W3] There is room for improvement in the numerical experiments to justify the linearization in the proposed method ([So2]).


**Soundness**

[So1] The derivation of GDN is theoretically sound and reasonable. That is, the architecture of GDN comes from estimating the gradient of the loss function by a single layer of GNN.

[So2] The ablation study (p.11) wrote that $K\geq 2$ resulted in unstable training (p.11. Section 5.1). I would like to clarify how the authors judged this.

[So3] This paper claimed the advantages of the proposed method as follows:
1. GDN can incorporate prior knowledge by giving an initial guess of the true graph as an initial value.
2. By using GNNs, GDN can make inference on graphs whose size is different from those used during training.

The first point was verified by the ablation study (P10), although I think this is a general property of L2O methods ([N3]).
The second point was verified in Figure 3. Indeed, it has been claimed that the increase in error is modest with respect to graph size. Intuitively, I agree with this claim. However, it would be desirable to provide objective metrics to judge whether this performance degradation is really modest, such as comparison with baselines.

[So4] Numerical experiments have compared GDN with many baselines. The experimental settings of the baselines are carefully designed to make the comparison fair. In addition, since the baseline methods are described in detail, their reproducibility is high.

[So5] The proposed method has not been compared with existing methods for estimating graph structure from node features, such as those mentioned on p.9, because it was difficult to compare with them (Section 5 Baselines). However, I wonder if we can do so in the task of estimating the true graph structure from node features. For the proposed method, we should construct an empirical variance-covariance matrix or the precision matrix and solve the problem of estimating the true graph structure from it, similar to the graph diffusion model.

**Clarity**

[C1] Most of the paper was clear. There was no great difficulty in understanding it. However, I think there is room for improvement in Related work in section 1 and Remark in section 3.

[C2] Although various concepts and existing methods were explained in the related work section, I felt that they were not well organized,
- The relationship between existing and proposed methods was only implicitly stated. For example, in the last part of the related work, some studies on latent graph learning and algorithm unrolling were mentioned. But we could not see why they were there in the first reading.
- Deep generative models have been mentioned. However, I think they are less relevant to the proposed method because they are not used in the supervised setting nor in graph recovery problems.

**Novelty and Significance**

[N1] In this paper, the authors considered the problem of estimating the true graph structure from an observed graph. This problem setting includes the problems considered in Pu et al. (2021) and Shrivastava et al. (2019). To the best of my knowledge, there are no known papers that apply GNNs to these problems. In this sense, this paper is novel.

[N2] The proposed method allows generalization to test data of different sizes, which was not possible with previous techniques. Therefore, I think this point is significant, although I think it would be desirable if we could verify it in a more objective way.

[N3] The fact that GDN can introduce prior knowledge as an initial guess has been mentioned as an advantage. However, I think this property is common to other L2O methods, although they are not applicable to the problem considered in this paper. Therefore, the novelty of the proposed method is limited from this point of view.

[N4] The technique of replacing the gradient of the objective function with a linearized model to reduce the computational complexity was a new technique in the context of L2O. On the other hand, as pointed out in [So2], it would be desirable to improve the validation of its effectiveness.

---

> ### Author Response · Authors · 2023-03-14
> **Authors’ response to Reviewer AruC (Part 1/2)**
>
> Thanks for your time and effort in reviewing our manuscript, as well as for finding key aspects of the paper novel and the derivation of GDN technically sound. We appreciate you acknowledging the comprehensiveness, detailed description and careful design of our experiments, which makes them highly reproducible. Overall, we are grateful for your thorough and clearly structured review, along with your valuable suggestions to improve the paper’s clarity. Point-by-point responses to your comments and associated requests for changes follow. We strive to improve our paper and will be happy to continue the discussion if any outstanding issues remain.
>
> &nbsp;
>
> **Soundness**
>
> [So2] The $K\geq 2$ case results in $\mathbf{A}^K$ terms in each GDN layer. These polynomial functions of the previous layer’s hidden representation imply the resulting unrolled network is *not* a neural network (namely, learnable *affine maps* followed by a pointwise nonlinear activation function). This divergence from neural network models is the main theoretical motivation for the adoption of $K=1$ via gradient truncation. As also mentioned in the `Ablation studies’ section, the practical motivation came from our observation that the training loss fluctuated greatly during training for $K\geq 2$ models (often causing numerical issues, e.g. NaN’s). We believe that the $K\geq 2$ model's optimization landscape is less benign as polynomial optimization is known to be notoriously hard, leading to less well-behaved gradients and consequently causing difficulties in training.
>
> [So3] This point is well taken. We agree that the possibility of leveraging prior information through the initial state $\mathbf{A}[0]$ of the unrolled architecture is not a unique feature of GDN. Indeed, it is a general property of unrolled architectures used for learning to optimize (L2O). This clarification notwithstanding, related methods such as GLAD or L2G have not thoroughly explored this additional flexibility in the initial state to incorporate prior information. For instance, L2G always utilizes an all-zero initialization but can accommodate complex learnable topological priors via an autoencoder module. A brief clarification along these lines has been included in the revised manuscript; please check the closing sentence of the `Incorporating prior information via algorithm initialization’ under Section 4.3
>
> [So5] We sincerely apologize, but we could not quite understand what you are proposing as the task to `estimate the true graph structure from node features’. Meaning, how does this differ from what we are actually doing in Section 5.1 - Learning graph structure from diffused signals? In that section, we start from node features $\mathbf{x}$ (adhering to a graph filter-based generative model) that we use to form a covariance matrix $\mathbf{A}_O=\mathbf{\Sigma}_x$. We then use a GDN to solve the problem of estimating  the true graph structure $\mathbf{A}_L$ from said covariance matrix. In particular, we compare against baseline methods to estimate graph structure from graph signals such as SpecTemp. We also compare against graph neural network (GNN)-based encoder-decoder architectures that could be used for link prediction. For these latter baselines, we use the covariance matrix as graph in the GNN encoders (which can be of two flavors: a graph convolutional network; and a graph isomorphism network), the nodal degrees as node features, and reconstruct the sought graph topology via an outer product decoder.
>
> &nbsp;
>
> **Clarity**
>
> [C2] Following your suggestions, we have carefully streamlined the presentation of the material in the `Related work’ section. Specifically, we removed the description of deep generative models for graphs that are only tangentially related to the graph recovery task dealt with here. We also clarified why existent latent graph learning and algorithm unrolling approaches are relevant. In particular, latent graph learning methods also use neural networks to predict adjacency matrices. Algorithm unrolling is the principle we adopt for architectural design in the novel graph structure learning setting dealt with here.

---

> > ### Author Response · Authors · 2023-03-14
> > **Authors’ response to Reviewer AruC (Part 2/2)**
> >
> > **Requested Changes**
> >
> > [R1] Yes, the novel supervised setting for topology identification refers to the case where we are in possession of a dataset of graph pairs, i.e., the observed $\mathbf{A}_{O}$’s and sparse latent $\mathbf{A}_L$’s. Specifically, what is novel here is the *network deconvolution problem setting* described in Section 2 - Problem Formulation. We seek to identify a sparse adjacency matrix $\mathbf{A}_L$ that encodes direct dependencies, when given an adjacency matrix $\mathbf{A}_O$ containing extraneous indirect relationships; see equation (1). The ‘Remark: Graph convolutional data model in context’ in Section 3 - Motivating Application Domains and the Supervised Setting shows how data model (1) relates to – and generalizes -- previous graph structure learning (a.k.a. network topology inference) data models. Some of these methods also operate in the supervised setting; for instance GLAD.
> >
> > [R2] That is certainly a valid interpretation of what we learn here.
> >
> > [R3] Yes, all such terms refer to the same task. Graph structure learning and latent graph learning are typically used by the machine learning on graphs community. Network topology inference has been coined in statistics. To avoid confusion and per your suggestion, in the revised paper we will adopt `graph structure learning’ throughout.
> >
> > [R4] The initial values of the block diagonal prior are given by the entries of the edge connection probability matrix used to sample the graph ensemble (that is, the stochastic block model parameters). This has been clarified in the revised manuscript; please check the `Ablation studies’ section.
> >
> > [R5] Yes, that is correct.
> >
> > [R6/8/9] Thanks for spotting these typos and inconsistencies in the notation. All fixed in the revised manuscript.
> >
> > [R7] The notation $\mathbf{H}(\mathbf{A}; \mathbf{h})$ seeks to emphasize that $\mathbf{A}$ is viewed here as an input, while $\mathbf{h}$ are parameters (hence after the semicolon, which is typical for models in the frequentist inference literature). We use the same convention for the GDN map $\Phi(\mathbf{A}_O;\bm{\Theta})$ to differentiate the role of inputs and learnable parameters. We are inclined to retain the current notation, but can change it to the simplified version $\mathbf{H}(\mathbf{A})$ if you still believe it will enhance clarity.
> >
> > &nbsp;
> >
> > Thanks again for your review.

---

> > > ### Comment · Reviewer_AruC · 2023-03-16
> > > **Response to authors' comments**
> > >
> > > I thank the authors for the detailed response. Here is the point-by-point answer to the response:
> > >
> > > [So2] Thank you for your clarification. I suggest writing the details of unstable training (e.g., fluctuation of training loss) and its hypothetical causes (e.g., hardness of optimization landscape.)
> > >
> > > [So3] OK. I agree that existing methods, such as GLAD or L2G, do not utilize the initial state for incorporating prior knowledge and that one of the strengths of this paper would be leveraging this property.
> > >
> > > [So5] I am sorry for confusing the authors. My question was whether it was justifiable that the proposed method was not compared with GNNs for link-prediction tasks (Wang et al., 2019; Kazietal., 2020; Veličković et al., 2020; Kipf et al., 2018; Kipf & Welling, 2016; Zhang & Chen, 2018). I think GNNs could apply to the experiment by computing the empirical covariance matrix.
> > >
> > > [C2] Thank you for revising the Related work section. This section becomes more precise than the previous version. Still, I suggest two points for improvement.
> > >
> > > 1.
> > > > Recent advances were [...] to signal model misspecifications. Scalability is an issue for the spectral-based network deconvolution approaches [...] for each problem instance.
> > >
> > > I think the connection between these two sentences can be improved. The first sentence does not talk about the spectral-based network deconvolution approaches. However, the second sentence mentioned the issue of this approach. Therefore, readers may feel an abrupt change of the topic occurred.
> > >
> > > 2.
> > > > When it comes to this latter objective, [...]
> > >
> > > I think it is unclear what this latter objective indicates.
> > >
> > > [R1--4, 6, 8, 9] OK
> > >
> > > [R5] OK. I suggest writing this procedure explicitly to reduce ambiguity.
> > >
> > > [R7] OK. I understand the authors' intention. It is OK to keep the notation for consistency with the GDN maps.

---

> > > > ### Author Response · Authors · 2023-03-17
> > > > **Thank you, suggestions implemented in the revised manuscript.**
> > > >
> > > > Thanks for engaging in the discussion and for your additional suggestions to improve the manuscript. We have implemented those in the revised paper, namely:
> > > >
> > > > - We further revised the 'Related work' section to better connect the noted sentences and enhance clarity;
> > > > - We have added additional details on the unstable training we faced when $K\geq 2$ and its hypothetical causes; please check the closing discussion in the `Ablation studies' section; and
> > > > - We have carefully spelled-out the procedure followed to generate our (pseudo-)synthetic datasets; please check the revised Appendix A.6.
> > > >
> > > > Thanks again for your feedback and we look forward to a favorable recommendation.

---

### Review · Reviewer_VMWN · 2023-03-06

**Summary Of Contributions:**

This paper studies the graph structure learning/recovery problem and it can have the potential to solve many real applications (more explanations are required). Given the core graph decomposition equation in Eq (1), the authors propose a Graph Deconvolution Network (GDN) algorithm. GDN utilizes an iterative optimization manner for learning and unrolling for inference.


**Audience:**

Yes

**Claims And Evidence:**

Yes

**Requested Changes:**

1. The writing can be improved. There is some key information missing in the draft, and some contents seem to be redundant (or more explanations on the difference are required). Please check the comments above.

2. The authors can polish the Introduction section, especially on the motivation and technical novelty of GDN. Current writing cannot reveal the importance of the problem and contributions of this work.

3. The equation references are missing. It should be “Eq/Equation (x)” instead of “(x)”.


**Strengths And Weaknesses:**

1. This paper is an empirical work, but the introduction lacks motivation/intuition. More concretely, the first two paragraphs list the algorithms or conclusions of the existing works, e.g., the graph structure can be further optimized for downstream tasks. However, the use cases of such algorithms are not well discussed. What are the real applications and corresponding impacts that we need to optimize the downstream graph structure? And how to interpret the modification to the graph structure? Adding such explanations can help the audience know better about the importance of this task. Currently, this information is only vaguely discussed in Introduction.

2. Also, in the Introduction, in the second paragraph, the authors mention they propose GDN, but what is the relation between GDN and related works? This is not explained in the Introduction.

3. In Sec 2, I’m wondering if the authors can explain what is the relation between E and A_O and A_L? Now the formulation is not straightforward without a detailed explanation/intuition.
- According to the context, I guess that E is equivalent to A_O, and they are known as inputs; A_L is the latent variable that we want to learn. Is this correct?
- Also, can authors also help explain what are the real scenarios corresponding to A_O and A_L? First, the draft says A_O contains “spurious, indirect connections”, but such a description is not very straightforward for readers. Then, the authors list some use cases in the last paragraph in Sec 2, but they are not self-contained and reader-friendly to audiences not familiar with the domain-specific field. I would recommend authors explain this into more detail, and a simple case can be something like what are the A_O and A_L for each application.
- The authors have some examples in Sec 3, I’m wondering what the relationship is between the examples in Sec 2 and 3.

---

> ### Author Response · Authors · 2023-03-15
> **Authors’ response to Reviewer VMWN (Part 1/2)**
>
> Thanks for your time and effort in reviewing our manuscript, as well as for recognizing the potential breadth of applicability of the proposed framework for supervised graph structure learning. We appreciate your valuable suggestions to improve the paper’s presentation and its clarity. Point-by-point responses to your comments and associated requests for changes follow. We strive to improve our paper and will be happy to continue the discussion if any outstanding issues remain.
>
> &nbsp;
>
> **Optimizing the Graph for a Downstream Task**
>
> When a downstream machine learning task is to be performed on a given graph $\mathbf{A}_O$, if possible it is often desirable to instead use a sparser graph $\mathbf{A}_L$ to boost performance, or, simply make the task tractable in big data scenarios. Examples include graph sparsification while preserving spectral properties, co-learning or ‘augmenting’ the graph used in a GNN for e.g., semantic segmentation of point cloud data, parametrizing graph-based regularization layer for CNNs, node classification in citation, social, and transportation networks, just to name a few; see e.g.,
>
> - D. A. Spielman and N. Srivastava, “Graph sparsification by effective resistances,” SIAM J. Comput., 2011 https://arxiv.org/pdf/0803.0929.pdf
> - A. Kazi et al, “Differentiable graph module (DGM) for graph convolutional networks,” IEEE Trans. Pattern Anal. Mach. Intell., 2023 https://arxiv.org/abs/2002.04999
> - J. Svoboda et al, “PeerNets: Exploiting peer wisdom against adversarial attacks,”, ICLR, 2019 https://arxiv.org/abs/1806.00088
> - T. Zhao et al, “Data augmentation for graph neural networks,” AAAI, 2021 https://arxiv.org/abs/2006.06830
>
> We do not pursue this task-driven use case in the present paper and rather explore other application domains outlined in Section 3 - Motivating Application Domains and the Supervised Setting. However, we still wanted to bring it up in the Introduction to convey potential broader impacts of GDNs. Clarifying details along these lines have been included in the revised manuscript; please check the opening paragraph in Section 1.
>
> &nbsp;
>
> **Intuition and real scenarios for $\mathbf{A}_O$ and $\mathbf{A}_L$**
>
> As you point out, Section 3 - Motivating Application Domains and the Supervised Setting as well as the paragraph directly preceding it is dedicated to clarify the use cases, real applications, and impacts this work encompasses. For instance and to be very concrete, in the problem of *inferring structural brain networks from functional MRI signals* the observed graphs $\mathbf{A}_O$ correspond to functional connectomes (FC, the covariance matrix of fMRI signals), and the latent graphs $\mathbf{A}_L$ correspond to sparse structural connectomes (SC). There is ample evidence that FC links tend to exist where there is no or little structural connection, which naturally motivates the polynomial model in eq. (1); see e.g.,
>
> - J. S Damoiseaux and M.l D Greicius, “Greater than the sum of its parts: A review of studies combining structural connectivity and resting-state functional connectivity,” Brain Struct. Func., 2009
> - H. Liang and H. Wang, “Structure-function network mapping and its assessment via persistent homology.” PLOS Comput. Biol,  2017.
>
>
> While given the limited page budget constraints we cannot go into detail of each of the applications we list, in the revised manuscript we explicitly indicated the latent graph $\mathbf{A}_L$ and observed graph $\mathbf{A}_O$ for the various examples mentioned. We want to stress that *we do provide detailed descriptions of the applications investigated in Section 5.2 - Real Data*. In closing we note that in most of these applications, $\mathbf{A}_L$’s will tend to be markedly sparser than $\mathbf{A}_O$’s; the extra edges in the former typically correspond to higher-order/indirect interactions between variables (i.e., nodes) which we aim to remove.

---

> > ### Author Response · Authors · 2023-03-15
> > **Authors’ response to Reviewer VMWN (Part 2/2)**
> >
> > **Relation of GDN to Related Works**
> >
> > The `Related work’ section offers a high-level overview of graph structure learning methods and the positioning of GDN in context. In particular, we briefly discuss L2G and GLAD along with their respective signal model assumptions (smoothness and Gaussianity, respectively) – both are unrolling-based graph structure learning methods just like GDN. Differences are discussed further in Section 4.2 - Learning to infer graphs via algorithm unrolling, particularly the fact that GDN is the first NN (namely, a composition of learnable *affine maps* followed by a pointwise nonlinear activation function) unrolling for graph structure learning and its favorable scaling properties. We include both methods as baselines; see also the performance comparisons in Section 5. Moreover, under  ‘Remark: Graph convolutional data model in context’ in Section 3, we discuss how our data model (1) subsumes several workhorse approaches for graph structure learning.
> >
> > Following reviewer suggestions, we have carefully streamlined the presentation of the material in the `Related work’ section. Specifically, we removed the description of deep generative models for graphs that are only tangentially related to the graph recovery task dealt with here. We also clarified why existent latent graph learning and algorithm unrolling approaches are relevant. In particular, latent graph learning methods also use deep learning to predict adjacency matrices. Algorithm unrolling is the principle we adopt for architectural design in the novel graph structure learning setting dealt with here.
> >
> > &nbsp;
> >
> > **Clarifying $\mathcal{E}$, $\mathbf{A}_O$, and $\mathbf{A}_L$**
> >
> > Apologies for the unintended confusion. All we say in the opening of Section 2 - Problem Formulation is that we consider undirected and weighted graphs $\mathcal{G}(\mathcal{V},\mathcal{E})$, with a common vertex set $\mathcal{V}$. Set $\mathcal{E}$ generically denotes the edges of the particular graph being considered, which we can encode using adjacency matrices. In particular, the graphs with adjacency matrices $\mathbf{A}_O$ and $\mathbf{A}_L$ will in general have different edges. We could formally denote them as $\mathcal{E}_O$ and $\mathcal{E}_L$, but we believe there is no need since the information is equivalently encoded in the respective adjacency matrices $\mathbf{A}_O$ and $\mathbf{A}_L$. Of course, the relation between these two adjacency matrices is given by the model (1).
> >
> > &nbsp;
> >
> > **Requested Changes**
> >
> > [R1/2] We have revised the opening paragraph in Section 1 - Introduction to include motivating use cases behind the idea of optimizing graph structure for downstream tasks. We have also streamlined the ‘Related work’ section to better position the technical contributions of GDNs. Moreover, we explicitly specify what $\mathbf{A}_O$ and $\mathbf{A}_L$ are in the application domains outlined at the end of Section 2 - Problem Statement. We will be happy to include any additional clarifications you deem relevant to improve the paper.
> >
> > [R3] Following your suggestion, we have modified the revised manuscript so that equations are referenced as eq. (x).
> >
> > &nbsp;
> >
> > Thanks again for your review.

---

### Decision · Action_Editors · 2023-05-09

**Recommendation:** Accept as is

**Comment:**

This paper studies the supervised graph structure learning/recovery problem. More specifically, the paper considers the problem of learning an adjacency matrix from a noisy input adjacency matrix A_O.  Then, the paper proposed Graph Deconvolution Network (GDN). This model is obtained by linearizing a single step of optimization by proximal gradient descent of an L1-regularized loss function.

Overall, the idea is solid and well supported by the numerical experiments; it is good to accept to TMLR.
Please prepare for a camera-ready version by including the author's name and updating the character colors.

**Audience:**

For Graph based machine learning problems, learning graph is an important topic, and TMLR's audience will be interested in the result.

**Claims And Evidence:**

The claims are well supported by numerical experiments.